# Geometrically Constrained Outlier Synthesis

**Daniil Karzanov** [1] [2]   **Marcin Detyniecki** [1] [3] [4]

## Abstract

Deep neural networks for image classification often exhibit overconfidence on out-of-distribution (OOD) samples. To address this, we introduce Geometrically Constrained Outlier Synthesis (GCOS), a training-time regularization framework aimed at improving OOD robustness during inference. GCOS addresses a limitation of prior synthesis methods by generating virtual outliers in the hidden feature space that respect the learned manifold structure of in-distribution (ID) data. The synthesis proceeds in two stages: (i) a dominant-variance subspace extracted from the training features identifies geometrically informed, off-manifold directions; (ii) a conformally-inspired shell, defined by the empirical quantiles of a non-conformity score from a calibration set, adaptively controls the synthesis magnitude to produce boundary samples. The shell ensures that generated outliers are neither trivially detectable nor indistinguishable from in-distribution data, facilitating smoother learning of robust features. This is combined with a contrastive regularization objective that promotes separability of ID and OOD samples in a chosen score space, such as Mahalanobis or energy-based. Experiments demonstrate that GCOS outperforms state-of-the-art methods using standard energy-based inference on near-OOD benchmarks, defined as tasks where outliers share the same semantic domain as in-distribution data. As an exploratory extension, the framework naturally transitions to conformal OOD inference, which translates uncertainty scores into statistically valid p-values and enables thresholds with formal error guarantees, providing a pathway toward more predictable and reliable OOD detection.

[1]AXA AI Research [2]EPFL, Lausanne, Switzerland [3]TRAIL, Sorbonne Université, Paris, France [4]Polish Academy of Science, IBS PAN, Warsaw, Poland. Correspondence to: Daniil Karzanov <daniil.karzanov@axa.com>.

*Proceedings of the 43rd International Conference on Machine Learning*, Seoul, South Korea. PMLR 306, 2026. Copyright 2026 by the author(s).

## 1. Introduction

Test-time out-of-distribution (OOD) detection is an important component of any machine learning model. For classification tasks, the ability to identify an input as an outlier rather than as an instance of one of the training set classes is crucial for robustly handling novel or unexpected inputs that inevitably arise outside the curated training environment. Classification models trained without explicit regularization for outliers can develop overconfident decision boundaries, where instances far from all classes may still be confidently assigned to the nearest class (Du et al., 2022; Vernekar et al., 2019).

To illustrate how uncertainty-aware objectives shape representations in practice, we begin by discussing Virtual Outlier Synthesis (VOS) (Du et al., 2022), which leads to more robust decision boundaries. VOS addresses the challenge of data imbalance in anomaly detection, particularly when anomalous examples are scarce or entirely absent during training. The core idea behind VOS is to artificially generate synthetic outliers, enabling the model to better differentiate between in-distribution (ID) and OOD patterns. Unlike traditional data augmentation, which typically relies on perturbations of existing samples or generation of OOD data in the data space (i.e. the space of input images), VOS models and samples from regions in the feature space that are statistically unlikely under the distribution of ID data. This approach helps simulate plausible yet diverse abnormal examples.

Among existing methods, our work is similar to VOS in its emphasis on OOD-aware representation learning. However, a fundamental limitation of the VOS-based approach is the assumption that outliers can be modeled as samples drawn from a simple distribution (e.g., Gaussian) outside the support of the normal data (Siddiqi et al., 2023). This simplification may fail to capture the complex and often non-Gaussian nature of real-world anomalies, which can exhibit structured or domain-specific characteristics (Tao et al., 2023b). Consequently, the synthesized outliers may not accurately reflect the true anomaly space, potentially leading to poor generalization. The effectiveness of VOS also depends on the geometry of the learned feature space. If the latent space does not adequately separate normal and abnormal regions, synthetic outliers may overlap with nor-

mal samples or fall into irrelevant regions, reducing their utility.

One of our key contributions is to replace the reliance on pre-defined parametric distributions (such as class-conditional Gaussians) for outlier modeling. Instead, we propose a geometric synthesis framework that generates outliers by probing low-variance subspaces of the learned feature manifold, as identified by Principal Component Analysis in the feature space. To control the difficulty of the synthetic outliers, we aim to generate meaningful examples for regularization that are neither too close to real data embeddings (making them inseparable), nor trivially obvious outliers that are too easy for the model to identify as OOD.

Another key limitation in existing OOD literature is the heavy focus on far-OOD benchmarks, where test data is semantically distant from the training domain (e.g., evaluating an animal classifier on industrial objects). While useful, these benchmarks overlook what we argue is the more critical challenge for robust AI: near-OOD detection, where models must separate fine-grained categories within the same super-class (e.g., unseen animal breeds). Such cases are more likely to trigger catastrophic failures in practice due to high feature-space similarity. Accordingly, alongside standard far-OOD benchmarks, our work places strong emphasis on evaluating methods against near-OOD tasks, where samples come from the same domain as in-distribution classes but remain unseen during training.

Conformal prediction (CP) (Vovk et al., 2005) addresses this challenge from a complementary perspective. CP provides a model-agnostic framework for quantifying uncertainty with formal statistical guarantees. It can be applied at inference time to any base predictor (Shafer & Vovk, 2008; Saunders et al., 1999; Vovk et al., 2017). Specifically, CP outputs a set or interval that contains the true label with guaranteed coverage. While OOD detection and CP originate from distinct lines of work, both ultimately aim to assess when a model's predictions should not be trusted. This motivates our exploration of whether CP can, **during training**, strengthen a model's robustness and its ability to internally flag outliers. We further investigate whether combining CP **during inference** offers a promising approach for more predictable and reliable OOD detection by translating uncertainty scores into statistically valid p-values with formal error guarantees.

The contributions of this paper can be summarized as follows: We first introduce a novel geometrically-driven outlier synthesis approach based on a conformal heuristic. Second, we propose a loss function incorporating nonconformity scores. Finally, we explore an alternative to energy-based inference as future work: conformal hypothesis testing for OOD detection.

Introducing our conformal heuristic into loss regulariza-

tion via outlier synthesis (first and second contributions) establishes a robust framework for OOD detection. Beyond this core method, we also provide an exploratory extension aimed at bridging the gap to provably reliable systems, by providing a formal statistical framework that governs their behavior under uncertainty. We discuss how this approach can be applied and, in doing so, open a new avenue for future research, with promising preliminary results on the Colored MNIST and Retinopathy datasets.

## 2. Background

### 2.1. Outlier Exposure

VOS regularization is based on the principle of Outlier Exposure (Hendrycks et al., 2019). During training, a set of virtual outliers, $\mathcal{Z}_{ood}$, are generated (e.g., by sampling from the tails of class-conditional Gaussian distributions fitted to in-distribution features). The OOD detection capability is learned by forcing a distinction between the energy scores (LeCun et al., 2007; Ngiam et al., 2011; Grathwohl et al., 2020) of ID and virtual outlier samples.

$$E(\mathbf{z}; \theta) = -\log \sum_{k=1}^{K} \exp(f_k(\mathbf{z}; \theta)) \qquad (1)$$

Following the formulation in the VOS paper, the energy score for a feature vector $\mathbf{z}$ with model parameters $\theta$ is defined as the negative log partition function (1) where $f_k(\mathbf{z}; \theta)$ is the $k$-th logit from the classifier for $K$ classes. A low energy score corresponds to a confident, in-distribution-like prediction. The model then learns a function $\phi$ that maps this energy score to a new logit, which determines the probability of the input being in-distribution. The uncertainty regularization loss, $\mathcal{L}_{\text{uncertainty}}$, is the binary cross-entropy loss for this task, where ID samples have a target label of 1 and virtual outliers have a target label of 0.

$$\begin{aligned} \mathcal{L}_{\text{uncertainty}} = {} & \mathbb{E}_{\mathbf{z}_{\text{in}} \sim \mathcal{Z}_{\text{in}}} \left[ -\log \frac{1}{1 + \exp(-\phi(E(\mathbf{z}_{\text{in}}; \theta)))} \right] \\ & + \mathbb{E}_{\mathbf{z}_{\text{ood}} \sim \mathcal{Z}_{\text{ood}}} \left[ -\log \frac{\exp(-\phi(E(\mathbf{z}_{\text{ood}}; \theta)))}{1 + \exp(-\phi(E(\mathbf{z}_{\text{ood}}; \theta)))} \right] \end{aligned} \qquad (2)$$

Here, $\mathcal{Z}_{\text{in}}$ is the distribution of features of the in-domain data and $\mathcal{Z}_{ood}$ is the distribution of synthesized virtual outliers. The first term pushes the probability of ID samples being recognized as ID towards 1, while the second term pushes the probability of virtual outliers being recognized as ID towards 0.

### 2.2. Energy-Based Inference

A common and widely adopted approach for OOD detection is the energy-based OOD detection score (Liu et al., 2020),

which uses the value in equation (1) directly as the outlier signal, bypassing any auxiliary heads. Higher energy indicates lower model confidence and thus a greater likelihood of OOD. Conceptually, this plays the same role as a probability score in binary classification, providing a per-sample measure of "outlierness." To ensure comparability with prior work and consistency with the VOS evaluation protocol, we report results in Section 5 using the general energy score as the OOD detection score.

While energy-based scores provide a simple heuristic, their thresholds are tuned on validation data and lack formal guarantees on novel inputs. In Appendix D, we explore an alternative approach that converts raw scores into statistically valid p-values, enabling thresholds with formally controlled error rates.

### 2.3. Conformal Prediction

Conformal prediction provides a distribution-free framework for constructing prediction sets with guaranteed coverage under the assumption that the data points are exchangeable. Let $\mathcal{D}_{\text{cal}} = \{(x_i, y_i)\}_{i=1}^n$ denote a calibration set drawn from the same distribution as the test points. For a new input $x_{n+1}$, a nonconformity score function $\mathcal{S}(x, \hat{y})$ measures how unusual a candidate label $\hat{y}$ is relative to the calibration data. The conformal prediction set for $x_{n+1}$ is then defined as $C(x_{n+1}) = \{\hat{y} : \mathcal{S}(x_{n+1}, \hat{y}) \leq Q_{1-\alpha}\}$, where $Q_{1-\alpha}$ is the $(1 - \alpha)$-quantile of the nonconformity scores on the calibration set. Under the exchangeability assumption, this guarantees that the prediction set covers the true label with probability at least $1 - \alpha$. The choice of nonconformity score is flexible and can be adapted to different types of models and tasks. For classification, it is often derived from model probabilities or logits; for regression, it can be based on residuals or prediction intervals.

Conformal prediction in the context of OOD detection can be formulated as hypothesis testing, where the nonconformity score functions as a test statistic. For a candidate test point, CP computes a p-value by comparing its score to the calibration scores from the training data, which is analogous to accepting or rejecting a null hypothesis (Vovk et al., 2014; Barber et al., 2023; Tibshirani et al., 2019). Similar to classical hypothesis testing, controlling Type I error (coverage guarantee) often comes at the cost of low power, i.e., the ability to correctly reject false hypotheses. Some OOD points may not appear "strange" enough to be flagged, particularly if the nonconformity score computed from the model's internal representations is weak. The nonconformity score derived from the model's final prediction $\hat{y}$ may not fully capture uncertainty. A single prediction must simultaneously serve as an effective uncertainty measure and as an accurate estimate of the true label or class probability, which can be a limiting constraint. In contrast, deriving the

score from logits or the model's hidden states provides more flexibility and often produces a more informative nonconformity measure. Consequently, in this work, we repurpose the conformal prediction framework as a geometric heuristic for training-time outlier synthesis. We note that this usage leverages the descriptive power of nonconformity scores to define decision boundaries, without claiming the statistical coverage guarantees which are strictly valid only under exchangeability conditions at inference time. Appendix D discusses how we can restore the guarantees with a post-hoc calibration.

## 3. Conformal Shell Synthesis

We introduce a geometrically-driven synthesis of virtual outliers via Principal Component Analysis (Hotelling, 1933). An eigen-decomposition is applied to the hidden representations $\mathcal{Z}_{\text{in}}$ obtained from the backbone network immediately preceding the classification layer. Once the eigenvectors and eigenvalues, $\mathbf{V}_{\text{train}}$ and $\mathbf{\Lambda}_{\text{train}}$, are computed, they are split into "large" and "small" components. The first $K$ principal components that account for at least $\eta$ (e.g. 90%) of the variance are marked as "large," while the remaining $\mathbf{V}_{\text{small, train}}^k$ are considered "small." These small PCs are the main building blocks for outlier synthesis: they correspond to directions with the smallest variability in the data, meaning that moving along them produces unlikely points while still remaining near the data centroid $\mu$. Similar to VOS, we use a rolling per-class buffer to compute the covariance matrix $\Sigma$ and the mean.

Let $v$ be the direction of a chosen small PC (see the scatter plot in figure 1). We can then generate an OOD feature as $\mathbf{z}_{\text{ood}}(\alpha) = \mu + \alpha v$, where $\alpha$ is a scalar. The problem of outlier synthesis is thus reduced to sampling a single scalar. Choosing $\alpha$ requires care: very small values produce synthetic points too close to real features to be separable, while very large values produce trivially easy outliers that are not useful for generalization.

To address this, we apply a conformal prediction-inspired heuristic to determine appropriate values for $\alpha$. This heuristic evaluates how "strange" a point is relative to a calibration set set aside at the beginning of training, ensuring that generated outliers avoid being overly simple, unrealistically extreme, or, at the other extreme, indistinguishably close to ID data. For a given choice of the score (uncertainty) function $\mathcal{S}$, we compute the quantiles of the scores of the features derived from the calibration data. These quantiles then serve as thresholds to define a conformal shell $[\alpha_{\text{inner}}, \alpha_{\text{outer}}]$. Intuitively, the conformal shell defines a range of deviation magnitudes, $\alpha$, that produce outliers of a specific "strangeness." The boundaries of this shell, $\alpha_{\text{inner}}$ and $\alpha_{\text{outer}}$, are determined by the 95th and 99th percentile nonconformity score thresholds ($q_{95}$ and $q_{99}$ respectively), which are derived from the

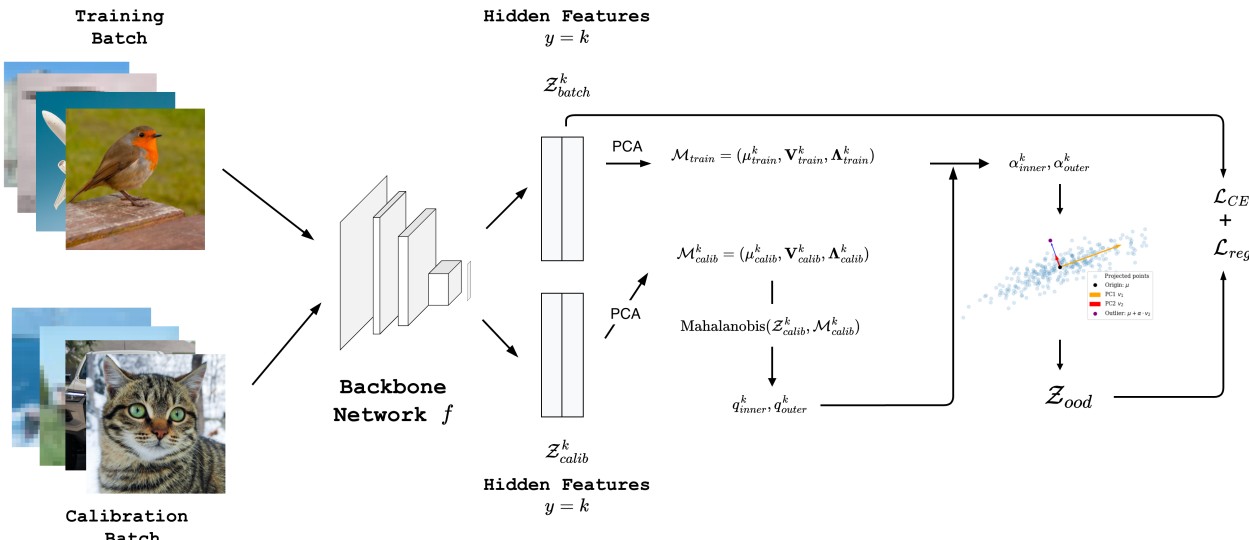

*Figure 1.* **GCOS Training Procedure Schematic.** Illustration of the data flow in our online synthesis and regularization method. Epoch-level calibration on $\mathcal{D}_{calib}$ produces class-conditional subspace models $\mathcal{M}_{calib}$ and Mahalanobis quantiles $q$. During batch-level training, features are used to update a queue that generates proposer subspace models $\mathcal{M}_{train}$ and identifies off-manifold directions $v$. Outliers $\mathbf{z}_{ood}$ are synthesized to match the target quantiles $q$, as evaluated by $\mathcal{M}_{calib}$. The final regularization loss, $\mathcal{L}_{reg}$, is a contrastive objective computed on the energy scores of in-distribution batch features and the synthesized outliers $\mathbf{z}_{ood}$ and added to cross-entropy loss, $\mathcal{L}_{CE}$, from the main classification task.

calibration set. These two percentiles correspond to the standard significance levels $\alpha = 0.05$ and $\alpha = 0.01$ in statistical hypothesis testing, and are used here as a principled default that does not require per-dataset tuning; sensitivity to the related variance threshold $\eta$ is examined in Appendix J. Formally, $\alpha_{\text{inner}}$ is the minimum deviation required such that the nonconformity score of the synthesized point equals the lower threshold, i.e., $S(\mathbf{z}_{ood}(\alpha_{\text{inner}})) = q_{95}$. This establishes $\alpha_{\text{inner}}$ as the precise boundary where, for any infinitesimal $\epsilon > 0$, the point $\mathbf{z}_{ood}(\alpha_{\text{inner}} - \epsilon)$ would still be considered in-distribution by this threshold, while $\mathbf{z}_{ood}(\alpha_{\text{inner}} + \epsilon)$ would not. Similarly, $\alpha_{\text{outer}}$ is determined using the higher score threshold $q_{99}$. New OOD features, $\mathbf{z}_{ood}(\alpha)$, are then generated by sampling $\alpha$ uniformly from this "hard-negative" shell: $\alpha \sim \mathcal{U}[\alpha_{\text{inner}}, \alpha_{\text{outer}}]$.

When choosing the direction $v$, we consider two options. Having identified the "small" eigenvectors $\mathbf{V}^k_{\text{small, train}}$, we either average them to obtain a single generation direction

$$v = \frac{1}{|\mathbf{V}^k_{\text{small, train}}|} \sum_{v_i \in \mathbf{V}^k_{\text{small, train}}} v_i \quad (3)$$

or we apply the outlier synthesis procedure separately for each $v_i \in \mathbf{V}^k_{\text{small, train}}$ or its subset. We refer to the former approach as the *average direction* method and the latter as the *per direction* method. Refer to Appendix J for the ablation study. Additionally, moving in the opposite direction of $v$ is also possible, which we implement by randomly selecting the sign of the direction.

Since the model is updated continuously and the calibration

data participates in this feedback loop, the strict exchangeability assumption of traditional conformal prediction is violated. To address this, we maintain two calibration sets: one for online calibration and outlier synthesis during training, and another for final calibration, which is later used for the conformal hypothesis testing during OOD evaluation in Appendix D.

If the score function is monotonic in $\alpha$ (i.e., uncertainty increases as we move farther from the centroid), $\alpha$ can be determined through single-variable optimization. Given the binary nature of OOD detection, where the goal is simply to determine if the score crosses a target quantile, a binary search can also be employed, as described in Algorithm 1.The Mahalanobis distance (Mahalanobis, 2018) in (4) satisfies this criterion.

$$\mathcal{S}_{Mahal}(z, \mu, \{\lambda_i\}, \{v_i\}) = \sum_{i=1}^{D} \frac{\left((z - \mu)^T v_i\right)^2}{\lambda_i + \epsilon} \quad (4)$$

To directly optimize the feature space for OOD separability, we propose a geometric regularization loss, $\mathcal{L}_{reg}$. This loss is formulated as a contrastive objective that minimizes the nonconformity scores of in-distribution samples while maximizing the scores of synthesized OOD samples. Let $\mathcal{S}_{\mathcal{L}}(\mathbf{z} \mid \mathcal{M}_k)$ denote the nonconformity score of a feature vector $\mathbf{z}$ with respect to the calibration model $\mathcal{M}_k$ for class $k$. The loss is defined as (5) where $y_{id}$ is the true class label for the ID virtual sample $\mathbf{z}_{id}$.

$$\mathcal{L}_{reg} = \mathbb{E}_{\substack{\mathbf{z}_{id} \sim \mathcal{Z}_{id} \\ \mathbf{z}_{ood} \sim \mathcal{Z}_{ood}}} \left[ \max(0, \mathcal{S}_{\mathcal{L}}(\mathbf{z}_{id}|\mathcal{M}_{y_{id}}) \right. \\ \left. - \min_{k} \mathcal{S}_{\mathcal{L}}(\mathbf{z}_{ood}|\mathcal{M}_k) + m) \right] \quad (5)$$

In equation (5), the positive term, $\mathcal{S}_{\mathcal{L}}(\mathbf{z}_{id}|\mathcal{M}_{y_{id}})$, represents the score of an in-distribution sample against its own class's reference model, which we aim to minimize. The negative term, $\min_k \mathcal{S}_{\mathcal{L}}(\mathbf{z}_{ood}|\mathcal{M}_k)$, is the score of a synthesized outlier against its best-fitting (i.e., minimum score) class model, which we aim to maximize. The nonconformity score function $\mathcal{S}_{\mathcal{L}}(\cdot)$ can be either the Mahalanobis distance, in which case $\mathcal{M}_k$ corresponds to the leading variance directions of the class-conditional feature distribution, parameterized by $(\mu_k, \mathbf{V}_k, \mathbf{\Lambda}_k)$, or the energy strangeness score (6), where $\mathcal{M}_k$ is implicitly the shared classifier head. The adaptive margin $m$ is set to the difference between the 95th and 50th percentiles of the positive scores in the batch, ensuring a dynamically scaled separation between ID and OOD score distributions. Refer to Appendix B for adaptive margin calculation.

Building on the general framework, we next present its instantiation in our proposed GCOS configuration. First, the set of virtual outliers $\mathcal{Z}_{ood}$ is generated using our geometric synthesis method where the conformal shell boundaries are determined by quantiles of the Mahalanobis distance score function. Second, for the loss in equation (5), we define the nonconformity score $\mathcal{S}_{\mathcal{L}}(\cdot)$ to be the Energy Strangeness Score:

$$\mathcal{S}_{\mathcal{L}}(\mathbf{z}) = \log \sum_{i=1}^{K} w_i \cdot \exp(h_\phi(\mathbf{z})_i) \quad (6)$$

Since this score is class-agnostic, the $\min_k \mathcal{S}_{\mathcal{L}}(\mathbf{z}_{ood}|\mathcal{M}_k)$ term in the loss simplifies to $\mathcal{S}_{\mathcal{L}}(\mathbf{z}_{ood})$. This hybrid approach leverages the geometric properties of the feature space to propose effective outlier locations, while directly optimizing the energy landscape which has shown to be a robust indicator for OOD detection (Du et al., 2022). We also evaluated our outlier synthesis approach using the regularization loss from VOS, and examined the case where both synthesis and loss regularization scores are given by the Mahalanobis distance (i.e., $\mathcal{S}_{\mathcal{L}}(\cdot) = \mathcal{S}(\cdot) = \mathcal{S}_{Mahal}(\cdot)$), detailed in Appendix H.

## 4. Experimental Setup

Many previous works have primarily focused on far-outliers, i.e., they evaluated models on OOD datasets that are unrelated to the training data (e.g., CIFAR-10 for ID and Textures for OOD). We argue that such scenarios are not realistic in practice. Therefore, we propose evaluating on

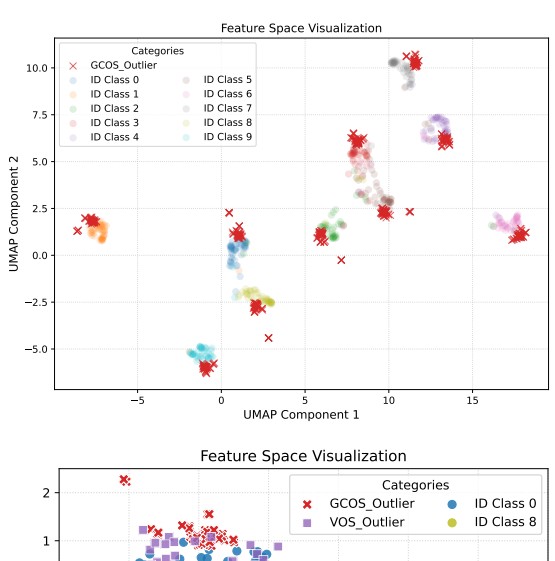

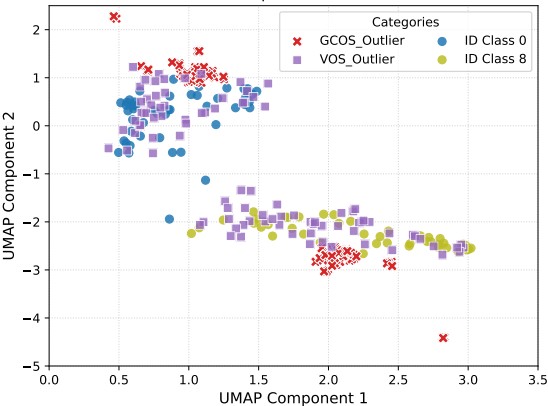

*Figure 2.* **UMAP Projection of Learned Features.** Top: the overall feature space, showing that classes form varying shapes and are largely well separated. Bottom: a zoomed-in view highlighting the distribution of GCOS outliers in off-manifold regions and VOS outliers near cluster edges for two clusters. The panels illustrate how GCOS generates points in challenging regions beyond the main clusters, while VOS outliers remain close to class boundaries.

near-OOD data - images that are not included in the training set but originate from the same field or domain. All main classification experiments use a WRN-40-2 backbone, matching the configuration of VOS (Du et al., 2022); the method itself operates on penultimate-layer features and is not tied to this specific architecture.

We consider four problems. The first dataset, *Colored MNIST*, assigns each digit in MNIST (LeCun, 1998) a fixed RGB color. The in-distribution split uses consistent digit-color associations, while the out-of-distribution split permutes these associations so that each digit appears with a different color. This construction yields an ID training set, an ID test set, and an OOD evaluation set, enabling controlled assessment of generalization under distribution shifts. The second dataset, *MVTec*, is based on images of defective industrial objects. From the original MVTec dataset (Bergmann et al., 2019), we construct the in-distribution training and test sets by splitting the "good", non-defective,

*Table 1.* OOD detection performance across four datasets. Best values per column shown in bold.

| Method | C-MNIST AUROC | AUPR | FPR95 | Dogs AUROC | AUPR | FPR95 | MVTec AUROC | AUPR | FPR95 | Retino AUROC | AUPR | FPR95 | Avg. AUC |
|---|---|---|---|---|---|---|---|---|---|---|---|---|---|
| No Reg. | 85.60 | 99.47 | 32.50 | 88.35 | 97.72 | 80.00 | 94.86 | 98.91 | 27.69 | 69.74 | 98.43 | 81.50 | 84.64 |
| VOS (Du et al., 2022) | 94.71 | 99.87 | 18.50 | 99.25 | 99.85 | 5.00 | 80.37 | 95.83 | 70.77 | 70.52 | 98.56 | 80.00 | 86.21 |
| MSP (Hendrycks & Gimpel, 2017) | 92.10 | 89.88 | 21.12 | 56.15 | 13.81 | 96.96 | 56.58 | 34.72 | 82.20 | 67.02 | 94.09 | 90.52 | 67.96 |
| MaxLogit (Hendrycks et al., 2022) | 88.88 | 85.68 | 27.59 | 58.42 | 14.27 | 93.69 | 55.30 | 39.20 | 90.24 | 77.21 | 96.41 | 81.39 | 69.95 |
| ReAct (Sun et al., 2021) | 89.41 | 86.60 | 24.69 | 57.77 | 14.08 | 94.04 | 57.31 | 36.74 | 87.09 | 77.37 | 96.43 | 75.84 | 70.47 |
| GradNorm (Huang et al., 2021) | 93.62 | 94.09 | 36.74 | 57.13 | 91.40 | 94.00 | 55.62 | 76.35 | 99.08 | 64.87 | 16.14 | 86.43 | 67.81 |
| KL-Matching (Hendrycks et al., 2022) | 91.96 | 89.70 | 21.41 | 56.39 | 13.82 | 93.11 | 56.29 | 34.55 | 83.62 | 69.00 | 94.48 | 84.28 | 68.41 |
| ViM (Wang et al., 2022) | 96.54 | 96.80 | 19.26 | 40.50 | 87.23 | 99.00 | 60.80 | 77.54 | 95.09 | 77.08 | 36.95 | 76.82 | 68.73 |
| Dream-OOD (Du et al., 2023) | 96.00 | 99.91 | 25.50 | 79.00 | 94.76 | 80.00 | 94.69 | 98.60 | 27.69 | 73.35 | 98.95 | 89.00 | 85.76 |
| NCIS (Doorenbos et al., 2024) | 96.72 | 99.93 | 24.50 | 99.35 | 99.87 | 10.00 | **96.50** | **99.09** | **3.08** | 75.29 | 99.05 | 85.50 | 91.97 |
| *GCOS (Ours)* | **99.50** | **99.99** | **1.00** | **99.55** | **99.91** | **0.00** | 95.61 | 99.08 | 23.08 | **79.23** | **99.16** | **73.00** | **93.47** |

samples of each class, while the OOD evaluation set consists of all defective samples from the same classes. This setup enables the evaluation of models on both normal variations and diverse anomalous patterns. The third dataset, based on *Stanford Dogs*, involves classifying multiple dog breeds (Khosla et al., 2011). In this case, the OOD data comprises breeds that are similar, but not identical, to those in the training set (e.g., golden retriever in the training set and labrador retriever as an OOD breed). Finally, the *Retinopathy* dataset (Karthik & Dane, 2019) consists of retinal fundus images curated for the classification of eye disorders. The in-distribution subset includes five diabetic retinopathy (DR) severity levels: No DR, Mild, Moderate, Severe, and Proliferative. To evaluate OOD detection, the dataset also provides fundus images with other ocular pathologies (Schwartz, 2020), such as glaucoma, age-related macular degeneration (AMD), and pathological myopia.

## 5. Results

### 5.1. Baseline Methods

We evaluate the trained models on the OOD detection task, where unseen ID test samples and OOD samples are mixed together, and the model is required to distinguish between them. We compare our approach against a model trained with Gaussian outlier synthesis with VOS regularization loss with an energy-based inference method, and a baseline model without regularization under energy-based inference. Additionally, we selected a representative set of baselines spanning different OOD detection paradigms. Among classical scoring methods, we utilize MSP (Hendrycks & Gimpel, 2017), which relies on the maximum softmax probability as a direct confidence proxy. Similarly, MaxLogit (Hendrycks et al., 2022) bypasses softmax normalization entirely, demonstrating that unnormalized logit magnitudes often carry more discriminative information regarding uncertainty than probability distributions. Moving to feature-space regularization, we evaluate ReAct (Sun et al., 2021), which truncates activations in the penultimate layer at a fixed threshold to suppress the abnormally high feature responses often triggered by OOD data. We also include ViM (Wang

et al., 2022), which explicitly models the ID manifold by combining standard logits with a residual score derived from the projection of features onto the null space of the principal components. In the gradient-based category, we use Grad-Norm (Huang et al., 2021), leveraging the observation that the vector norm of gradients backpropagated from the output layer is consistently higher for in-distribution inputs. We further represent distribution matching techniques with KL-Matching (Hendrycks et al., 2022), which detects anomalies by minimizing the Kullback-Leibler divergence between a sample's prediction and the mean class-conditional distributions. Finally, we compare against two SOTA methods for OOD detection. Dream-OOD (Du et al., 2023) pioneers the use of generative diffusion models for out-of-distribution detection by perturbing text embeddings to synthesize artificial outliers near the class boundaries. NCIS (Doorenbos et al., 2024) extends Dream-OOD by replacing the simple linear perturbation strategy with a normalizing flow-based mechanism (cVPN) that synthesizes outliers by modeling the complex, non-linear manifold of the in-distribution data. For our experiments, recent diffusion-based methods were adapted per dataset through focused hyperparameter tuning to ensure fair comparison.

For evaluation, we report the area under the ROC curve (AUROC; higher is better), the area under the precision-recall curve (AUPR; higher is better), and the false positive rate at 95% true positive rate (FPR95; lower is better). The first two metrics are threshold-agnostic, whereas the latter is defined for a threshold $\gamma$ such that 95% of in-distribution images are correctly identified as ID.

### 5.2. Discussion

As shown in Table 1, our approach - conformally driven outlier exposure with geometric regularization loss under energy-based inference - outperforms competing methods in terms of AUROC and AUPR. Moreover, it consistently achieves substantially lower FPR95 than VOS and no-regularization across all datasets (23.08% *vs.* 27.69% and 70.77%; 1% *vs.* 32.5% and 18.5%; $< 0.01\%$ *vs.* 80% and 5%; 73% *vs.* 81.50% and 80%). Interestingly, VOS reduces

the false positive rate compared to no-regularization on the Colored MNIST dataset; however, the trend reverses on the MVTec dataset, where the model without outlier exposure performs better than VOS. Nevertheless, our approach demonstrates superior and consistent performance across both datasets.

GCOS substantially outperforms a diverse set of baseline methods spanning classical scoring approaches, post-hoc regularization techniques, gradient-based methods, distribution matching, and non-parametric outlier synthesis. GCOS demonstrates state-of-the-art performance with the highest average AUROC of 93.47%, outperforming modern competitive methods such as NCIS (91.97%) and Dream-OOD (85.76%). While NCIS achieves slightly better performance on MVTec, our approach is more lightweight and scalable, avoiding the computational cost of diffusion model inference during training. The consistent superiority across diverse datasets suggests that geometry-aware outlier synthesis with conformal calibration provides fundamental advantages over both classical scoring methods that operate on fixed representations and synthesis methods that lack geometric guidance. The particularly large gaps on datasets with complex class distributions (such as Colored MNIST) indicate that adaptive per-class calibration is crucial when class geometries are heterogeneous, while the more modest improvements on MVTec suggest these datasets may have simpler decision boundaries where standard methods already perform near-optimally.

We decided to visualize the resulting feature space of our model using a UMAP projection (McInnes et al., 2018) in figure 2. We observe that our method separates most of the classes effectively. The shapes of the resulting "clusters" are not uniform; some appear more spherical while others are elongated. This may indicate that class-conditional Gaussian tail sampling does not fully capture the complex, low-dimensional manifold structure of learned feature representations, and that methods leveraging the geometry of the feature space are necessary. Another interesting property of the obtained representations is the behavior of GCOS outliers: they tend to cluster on the opposite sides of two nearby classes. This can be clearly seen in the bottom panel of figure 2, where VOS outliers are distributed around the cluster, including regions between classes where the decision boundary lies, whereas GCOS outlier synthesis behaves as intended. Rather than attempting to affect the main classification boundary, it instead pushes the decision boundary beyond both the cluster and the classification region closer to the data. As noted before if an outlier lies far beyond both the classification boundary between clusters and one of the clusters, models trained without OOD regularization will be confidently misclassifying such points as belonging to the adjacent class. By focusing on off-manifold directions, our method generates points in these challenging regions,

thereby flanking the decision boundary more tightly around the data clusters.

To summarize, GCOS improves OOD detection by generating outliers that respect the geometry of in-distribution data, consistently outperforming baseline methods across multiple metrics. Feature-space visualizations show that GCOS tightly enclose the decision boundary around data clusters, reducing overconfidence on challenging outliers. These results highlight the value of geometry-aware outlier synthesis for robust neural networks and point toward future integration with formal conformal guarantees.

## 6. Conclusion

We introduced GCOS, a training-time outlier-synthesis framework that combines class-conditional low-variance PCA directions with a conformally-calibrated shell to place synthetic outliers in a controlled hard-negative regime. Under standard energy-based inference, GCOS consistently improves near-OOD detection over Gaussian and diffusion-based synthesis baselines, operating entirely in the penultimate feature space and avoiding the cost of image-space generation. Overall, the experiments support the view that geometry-aware, calibration-driven synthesis is a useful complement to existing training-time OOD methods.

## Impact Statement

This paper aims to improve the reliability of deep classifiers by helping them recognise inputs outside the training distribution. Better OOD detection is especially useful in safety-critical settings such as medical imaging and industrial inspection, where confidently misclassifying unfamiliar inputs can lead to harmful decisions. Our Retinopathy experiments illustrate this regime. We do not foresee specific negative societal impacts beyond those of deep classifiers generally: GCOS modifies only the training objective and introduces no new data sources, generated images, or surveillance capabilities. As with any OOD method, practitioners should validate performance on deployment-representative data before relying on it in safety-critical decisions, since guarantees outside the calibration regime remain heuristic.

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

# Appendix

## A. Related Work

**Out-of-distribution detection** has been widely studied across several related tasks, including anomaly detection, novelty detection, and open-set recognition. Yang et al. (2024) provide a unified framework that situates these tasks as special cases, reviewing classification-, density-, and distance-based approaches and highlighting challenges such as calibration and realistic outlier modeling.

A common strategy for improving OOD detection is to leverage auxiliary outlier data. Hendrycks et al. (2019) introduced Outlier Exposure, showing that training on auxiliary outliers improves detection of unseen anomalies. Wang et al. (2023) extend this idea by crafting a set of worst-case OOD distributions around auxiliary data, theoretically reducing the discrepancy to unseen OOD samples. Li & Zhang (2025) propose synthesizing virtual outliers directly from in-distribution data using Hamiltonian Monte Carlo, generating diverse OOD samples without requiring an external dataset. Ming et al. (2022) also focus on informative outlier selection, using posterior sampling to learn a compact decision boundary between ID and OOD data.

Tao et al. (2023a) introduced Non-Parametric Outlier Synthesis (NPOS), which relaxes the restrictive Gaussian assumption of prior works by employing a non-parametric framework to synthesize outliers via rejection sampling. Addressing the challenge of high-dimensional generation, Du et al. (2023) proposed Dream-OOD, utilizing diffusion models to generate photo-realistic outliers in pixel space by sampling from low-likelihood regions of a text-conditioned latent space. Similarly, Liao et al. (2025) developed BOOD, a framework that specifically targets the decision boundary by perturbing in-distribution features to cross into OOD regions before decoding them into images using diffusion models. Doorenbos et al. (2024) presented NCIS to further enhance synthetic outlier quality, operating directly within a diffusion model's embedding space and employing a conditional volume-preserving network to model complex class manifolds. Finally, in the specific context of industrial anomaly detection, Roth et al. (2022) proposed PatchCore, which uses a memory bank of nominal patch-level features to achieve state-of-the-art performance on benchmarks like MVTec AD without requiring outlier supervision.

Other methods focus on feature-space metrics or model activations. Lee et al. (2018b) use Mahalanobis distance to define a confidence score, while Sun et al. (2022) employ non-parametric nearest-neighbor distances for flexible, assumption-free detection. Hsu et al. (2020) decompose confidence scoring and modify input preprocessing to improve detection without relying on OOD data. Sun et al. (2021) show that controlling internal activations can mitigate overconfidence on OOD samples. Several works address scalability and structured label spaces. Huang & Li (2021) propose MOS scoring, grouping labels to simplify decision boundaries and reduce false positives. Hendrycks & Gimpel (2017) establish the maximum softmax probability as a competitive baseline, and Du et al. (2024) demonstrate that unlabeled data can be leveraged to identify candidate outliers and train OOD classifiers with theoretical guarantees. Test-time adaptation has been explored as a dynamic approach to OOD detection. Yang et al. (2023) propose AUTO, which selectively updates the model using streaming pseudo-labeled in/out samples to improve robustness without requiring auxiliary OOD data.

**Conformal prediction** (Vovk et al., 2005) has emerged as a practical and widely applicable framework for uncertainty quantification. Angelopoulos & Bates (2022) provide an accessible overview of the methodology, emphasizing its distribution-free validity guarantees, ease of use, and adaptability to diverse domains such as computer vision, natural language processing, and reinforcement learning. Tibshirani et al. (2019) extend the approach to handle covariate shift by introducing a weighted version of conformal prediction that incorporates likelihood ratios between training and test distributions, thereby enabling valid prediction intervals even when data distributions differ. The interplay between conformal prediction and OOD detection has been explored, but to a limited extent. Novello et al. (2024) show that conformal methods can provide conservative corrections to OOD evaluation metrics, while OOD scores can improve conformal prediction sets when used as nonconformity measures. Finally, the theoretical scope of conformal prediction has been expanded substantially, as Prinster et al. (2024) prove that validity guarantees hold under any data distribution, including sequential feedback settings, and propose tractable algorithms for scenarios such as black-box optimization and active learning.

## B. Model Training

Algorithm 3 summarizes the end-to-end training procedure of our proposed GCOS framework. The process consists of two phases embedded in the standard training loop: an epoch-level online calibration step and a batch-level synthesis and regularization step.

**Epoch-Level Online Calibration.** The purpose of this step is to establish a stable "Judge" model that guides synthesis throughout the epoch. At the beginning of each epoch (after a warm-up period $E_{start}$), the model is evaluated on the held-out calibration set $\mathcal{D}_{calib}$ to obtain feature representations $\mathcal{Z}_{calib}$ for all in-distribution classes. For each class, a class-conditional linear subspace model $\mathcal{M}_{calib}^k$ is fit to its features, and the resulting Mahalanobis score dis-

tribution $S_{calib}^{Mahal,k}$ is computed. From this distribution, the quantiles $q_{inner}^k$ and $q_{outer}^k$ are extracted and stored as synthesis boundaries for the upcoming training iterations. For correctness of the subspace estimation, the data should be normalized across all dimensions so that dimensions with larger magnitudes do not bias the resulting components. We opt for a linear formulation (Hotelling, 1933) rather than a kernelized variant (Schölkopf et al., 1997) because we work in the feature space immediately before the final classification layer, where the data should already be linearly separable by the main classification head.

**Batch-Level Training.** Each batch begins with a standard forward pass and cross-entropy loss computation. The extracted features $Z_{id}$ are stored in a running feature buffer $\mathcal{Q}_{train}$, which serves as a "Proposer" by maintaining an estimate of the recent feature distribution. When the buffer is filled and the warm-up phase is complete, the GCOS synthesis and regularization procedure is applied. A Proposer PCA model $\mathcal{M}_{train}$ is constructed from $\mathcal{Q}_{train}$, and synthetic outliers are generated on a per-class basis. Specifically, for each class, we identify the low-variance principal components $\mathbf{V}_{small,train}^k$. Along these directions, the FindBoundaryAlpha subroutine determines the $\alpha$ values that place a candidate feature on the pre-computed quantile shells $q_{inner}^k$ and $q_{outer}^k$. Outliers $\mathbf{z}_{ood}$ are then sampled from these shells as perturbations of the class mean $\mu_{train}^k$.

After the synthesis step, the regularization loss is computed. In GCOS-Hybrid (Mahalanobis score for synthesis, energy-based score for regularisation loss), this involves calculating energy-based strangeness scores for both the real ID features ($S_{pos}$) and the synthesized outliers ($S_{neg}$). An adaptive margin, determined from the batch statistics of $S_{pos}$, is used to construct the contrastive regularization loss $\mathcal{L}_{reg}$. The final training loss combines the cross-entropy objective with the weighted regularization term, $\lambda\mathcal{L}_{reg}$, which is minimized through backpropagation to update the model.

## C. Final Post-Hoc Calibration for Inference

After training is complete, we perform a final, one-time calibration to prepare the model for inference. It is crucial to distinguish this final step from the "online" calibration performed during training. The online calibration uses the network's transient state at the start of each epoch to create an adaptive heuristic for guiding outlier synthesis.

In contrast, this final calibration operates on the fixed, fully-trained model and on another held-out calibration set. We perform a full forward pass of the calibration data, $\mathcal{D}_{calib}$, through this final model to generate a permanent reference distribution of nonconformity scores, denoted $\mathcal{S}_{calib}^{final}$. Crucially, this final distribution is distinct from the ephemeral, epoch-dependent distributions, $\mathcal{S}_{calib}^{(e)}$, used during training

---

**Algorithm 1** FindBoundaryAlpha

**Require:** Start point $\boldsymbol{\mu}_{train}$, direction $\mathbf{v}_j$, target score $q_{target}$, score function $\mathcal{S}$, search range $\alpha_{max}$, steps $N_{steps}$
**Ensure:** Boundary deviation $\alpha_{boundary}$
1: $s_0 \leftarrow \mathcal{S}(\boldsymbol{\mu}_{train})$
2: **if** $s_0 \geq q_{target}$ **then**
3: $\quad$ **return** 0
4: **end if**
5: $s_{max} \leftarrow \mathcal{S}(\boldsymbol{\mu}_{train} + \alpha_{max} \cdot \mathbf{v}_j)$
6: **if** $s_{max} < q_{target}$ **then**
7: $\quad$ **return** $\alpha_{max}$
8: **end if**
9: $\alpha_{low} \leftarrow 0, \alpha_{high} \leftarrow \alpha_{max}$
10: **for** $i = 1$ to $N_{steps}$ **do**
11: $\quad \alpha_{mid} \leftarrow (\alpha_{low} + \alpha_{high})/2$
12: $\quad \mathbf{z}_{cand} \leftarrow \boldsymbol{\mu}_{train} + \alpha_{mid} \cdot \mathbf{v}_j$
13: $\quad s_{cand} \leftarrow \mathcal{S}(\mathbf{z}_{cand})$
14: $\quad$ **if** $s_{cand} < q_{target}$ **then**
15: $\quad\quad \alpha_{low} \leftarrow \alpha_{mid}$
16: $\quad$ **else**
17: $\quad\quad \alpha_{high} \leftarrow \alpha_{mid}$
18: $\quad$ **end if**
19: **end for**
20: **return** $\alpha_{high}$

---

**Algorithm 2** AdaptiveMargin

**Require:** Set of positive scores $\mathcal{S}_{pos}$
**Require:** Low percentile $p_{low}$, high percentile $p_{high}$
**Require:** Default margin $m_{default}$
**Ensure:** Margin $m$
1: **if** $|\mathcal{S}_{pos}| > 1$ **then**
2: $\quad q_{low} \leftarrow p_{low}/100.0$
3: $\quad q_{high} \leftarrow p_{high}/100.0$
4: $\quad S_{typical} \leftarrow \text{Quantile}(\mathcal{S}_{pos}, q_{low})$
5: $\quad S_{boundary} \leftarrow \text{Quantile}(\mathcal{S}_{pos}, q_{high})$
6: $\quad m \leftarrow \max(0, S_{boundary} - S_{typical})$
7: **else**
8: $\quad m \leftarrow m_{default}$
9: **end if**
10: **Return** $m$

---

(i.e., $\mathcal{S}_{calib}^{final} \neq \mathcal{S}_{calib}^{(e)}$ for any epoch $e$ before training completion). Because the model is now a fixed function, this final calibration step satisfies the exchangeability assumption of conformal prediction. This allows $\mathcal{S}_{calib}^{final}$ to be used for generating statistically valid p-values for OOD hypothesis testing on unseen data, a formal guarantee the online heuristic cannot provide.

## D. Moving Towards OOD Detection with Statistical Guarantees

This appendix presents an exploratory extension of our framework along with preliminary results for future work. The conformal hypothesis testing head described below is an alternative inference procedure applied to the same trained models on the same datasets, so the natural point of comparison is Table 1 rather than external OOD baselines. At inference, the goal of OOD detection is to decide whether an unseen sample comes from the in-distribution or not. Traditional approaches assign a score and compare it against a threshold tuned on validation data. While often effective, such heuristic thresholding provides no formal guarantee of

---

**Algorithm 3** GCOS Training Procedure

---

**Require:** Training data $\mathcal{D}_{train}$, Calibration data $\mathcal{D}_{calib}$, Network $(f_\theta, h_\phi)$ (backbone $f$, classifier head $h$),
   $E$ epochs, start epoch $E_{start}$, loss weight $\lambda$, shell percentiles $(p_{inner}, p_{outer})$
1: Initialize network parameters $\theta, \phi$
2: Initialize feature queues $\mathcal{Q}_{train} \leftarrow \emptyset$
3: **for** epoch $e = 1$ to $E$ **do**
4:   **if** $e \geq E_{start}$ and $\mathcal{D}_{calib}$ is available **then**
5:     # Epoch-Level Online Calibration (The "Judge")
6:     Set network to eval mode
7:     **for** each class $k$ **do**
8:       Extract features $\mathcal{Z}^k_{calib}$ from $\mathcal{D}_{calib}$ using $f_\theta$
9:       Compute $\mathcal{M}^k_{calib} = (\mu^k_{calib}, \mathbf{V}^k_{calib}, \mathbf{\Lambda}^k_{calib}) \leftarrow \text{PCA}(\mathcal{Z}^k_{calib})$
10:      Compute Mahalanobis scores $S^{Mahal,k}_{calib} \leftarrow$ Mahalanobis$(\mathcal{Z}^k_{calib}, \mathcal{M}^k_{calib})$
11:      Compute thresholds $q^k_{inner}, q^k_{outer} \leftarrow$ Quantiles$(S^{Mahal,k}_{calib}, p_{inner}, p_{outer})$
12:    **end for**
13:    Set network to train mode
14:  **end if**
15:  **for** batch $(X, Y) \in \mathcal{D}_{train}$ **do**
16:    $Z_{id} \leftarrow f_\theta(X); L \leftarrow h_\phi(Z_{id})$
17:    $\mathcal{L}_{CE} \leftarrow \text{CrossEntropy}(L, Y)$
18:    Update $\mathcal{Q}_{train}$ with $Z_{id}$
19:    $\mathcal{L}_{reg} \leftarrow 0$
20:    **if** $\mathcal{Q}_{train}$ is full and $e \geq E_{start}$ **then**
21:      # GCOS Synthesis (Geometric Proposer)
22:      Compute $\mathcal{M}_{train} = (\mu_{train}, \mathbf{V}_{train}, \mathbf{\Lambda}_{train})$ from $\mathcal{Q}_{train}$
23:      $\mathcal{Z}_{ood} \leftarrow \emptyset$
24:      **for** each class $k$ **do**
25:        Identify small components $\mathbf{V}^k_{small,train}$ from $\mathbf{V}^k_{train}$ based on variance explained threshold $\eta$
26:        **for** direction $\mathbf{v}_j \in \mathbf{V}^k_{small,train}$ **do**
27:          # Algorithm 1.
28:          $\alpha_{inner} \leftarrow \text{FindBoundaryAlpha}(\mu_{train}, \mathbf{v}_j, q^k_{inner}, \mathcal{S})$
29:          $\alpha_{outer} \leftarrow \text{FindBoundaryAlpha}(\mu_{train}, \mathbf{v}_j, q^k_{outer}, \mathcal{S})$
30:          Sample $\alpha \sim \text{U}[\alpha_{inner}, \alpha_{outer}]$
31:          $\mathbf{z}_{ood} \leftarrow \mu^k_{train} + \text{sign} \cdot \alpha \cdot \mathbf{v}_j$
32:          $\mathcal{Z}_{ood} \leftarrow \mathcal{Z}_{ood} \cup \{\mathbf{z}_{ood}\}$
33:        **end for**
34:      **end for**
35:      # Regularization Loss (Energy-Based)
36:      Compute Energy strangeness scores: $S_{pos} \leftarrow \text{logsumexp}(h_\phi(Z_{id}))$, $S_{neg} \leftarrow \text{logsumexp}(h_\phi(\mathcal{Z}_{ood}))$
37:      $m \leftarrow \text{AdaptiveMargin}(S_{pos})$
38:      $\mathcal{L}_{reg} \leftarrow \text{Mean}(\text{ReLU}(S_{pos} - S_{neg} + m))$
39:    **end if**
40:    $\mathcal{L}_{total} \leftarrow \mathcal{L}_{CE} + \lambda \mathcal{L}_{reg}$
41:    Update $\theta, \phi$ using gradients of $\mathcal{L}_{total}$
42:  **end for**
43: **end for**
**Ensure:** Trained network $(f_\theta, h_\phi)$

---

performance on unseen data.

**Inference via Conformal Hypothesis Testing.** OOD detection using conformal inference provides a principled way to assess how unusual a new data point is, translating raw scores into statistically meaningful p-values and enabling formal hypothesis testing. In contrast, simple energy-based thresholding relies on heuristics - a cutoff tuned on past data that may fail when conditions change. By offering thresholds with rigorous error guarantees, conformal inference ensures that decisions remain reliable even on unseen data.

Knowing the uncertainty measure in advance may help guide the model's hidden representations to facilitate

inference-time conformal prediction, either during training or post-training. Therefore, our work integrates procedures from conformal inference into the learning process of deep learning models, illustrated on a computer vision classification task with OOD detection. We believe that this idea could represent a promising new direction in research. The manual choice of the nonconformity score function is reminiscent of handcrafted filters in the early days of computer vision (Szeliski, 2022; LeCun et al., 2015), which were later largely replaced by learned convolutional layers. In a similar spirit, recent methods have explored ways to integrate uncertainty-aware mechanisms directly into the learning process (Kendall & Gal, 2017; Lakshminarayanan et al., 2017; Gal & Ghahramani, 2016; Kuleshov et al., 2018). After training concludes, we run a single calibration step on a separate held-out calibration set to prepare the model for inference. This final calibration is distinct from the online calibration performed during training, which relies on the network's temporary state at the start of each epoch to build an adaptive heuristic for guiding outlier synthesis. Further details are provided in Appendix C.

**Empirical Analysis of Conformal Guarantees.** During inference, we attempt to apply a formal hypothesis testing framework based on conformal prediction to determine whether an unseen sample ID or OOD. This procedure replaces the energy-based inference in (1), where OOD detection relies on a computing energy score from the logits of the main classification head.

Given a test sample with feature vector $\mathbf{z}^{\text{test}}$, we first compute its nonconformity score, $S(\mathbf{z}^{\text{test}})$, using the score function obtained during final calibration (e.g., Energy or Mahalanobis). This score is then used to derive a p-value, which quantifies the likelihood that $\mathbf{z}^{\text{test}}$ belongs to the ID distribution under the null hypothesis.

To account for the multi-class nature of the ID data, we calculate a p-value for the test sample with respect to each class's reference distribution. For class $k$, this p-value is the fraction of samples in the final calibration set $\mathcal{S}^{\text{final}}_{calib,k}$ whose scores are greater than or equal to the test sample's score:

$$p_k(\mathbf{z}^{\text{test}}) = \frac{1 + |\{\mathbf{s} \in \mathcal{S}^{\text{final}}_{calib,k} \mid \mathbf{s} \geq S(\mathbf{z}^{\text{test}}|\mathcal{M}_k)\}|}{1 + |\mathcal{S}^{\text{final}}_{calib,k}|} \quad (7)$$

The overall p-value for the test sample is the maximum across all classes, $p_{\text{final}}(\mathbf{z}^{\text{test}}) = \max_{k \in \{1,...,K\}} p_k(\mathbf{z}^{\text{test}})$, reflecting its compatibility with the most likely ID class. A sample is classified as OOD if this final p-value is less than the significance level, and as ID otherwise.

We can observe from Table 2 that the use of conformal hypothesis testing as an OOD detection head, instead of energy-based inference discussed in section 2.2, yields

*Table 2.* Results using formal hypothesis testing (7) as OOD classification head.

| | MVTec | | | Colored MNIST | | | Stanford Dogs | | | Retinopathy | | |
|---|---|---|---|---|---|---|---|---|---|---|---|---|
| | AUROC | AUPR | FPR95 | AUROC | AUPR | FPR95 | AUROC | AUPR | FPR95 | AUROC | AUPR | FPR95 |
| GCOS + $\mathcal{L}_{\text{uncertainty}}$ | 50.92 | 83.68 | 100.00 | 98.75 | 99.97 | 2.00 | 50.00 | 83.33 | 100.00 | 67.60 | 98.37 | 77.00 |
| GCOS + $\mathcal{L}_{reg}$ (Mahalanobis) | 63.67 | 88.07 | 100.00 | 98.93 | 99.98 | 1.50 | 50.00 | 83.33 | 100.00 | 37.58 | 96.27 | 97.00 |
| GCOS + $\mathcal{L}_{reg}$ (Energy) | 70.62 | 88.50 | 72.31 | 76.00 | 97.85 | 34.00 | 2.40 | 65.43 | 100.00 | 31.11 | 95.30 | 98.00 |

mixed results. In some cases, this technique collapses to a nearly random classifier. On the simple Colored MNIST dataset, it achieves competitive scores, with an outstanding 1.5% FPR95 when applying geometric score-based regularization loss using the Mahalanobis distance. Interestingly, geometric loss with energy shows performance improvement on the MVTec dataset, reaching an AUROC of approximately 70.62%. For the Retinopathy dataset, our outlier synthesis combined with the logistic regression uncertainty loss from (2) (GCOS + $\mathcal{L}_{\text{uncertainty}}$) achieves comparable AUROC performance to VOS and the model without regularization evaluated under energy-based inference and reported in Table 1, while also improving FPR95. A plausible explanation for the collapse on Stanford Dogs and MVTec is calibration-set resolution: these datasets contain only 1,394 and 1,319 training images respectively (Appendix Table 3), so after the held-out conformal calibration split the per-class calibration sets become small and the resulting empirical $p$-values are coarse. By contrast, Colored MNIST provides roughly 600 calibration samples per class for the same split, and the conformal head performs strongly there. This pattern is consistent with a calibration-size bottleneck rather than a representation failure, since the same backbone yields substantially stronger results under energy-based inference (Table 1).

Despite mixed performance, these results come with statistical guarantees. At present, the technique demonstrates promising potential and represents an emerging area for further research. In the main results section, we demonstrated that incorporating conformal-inspired techniques improve the model's accuracy on OOD detection, representing a step toward a more advanced framework. If the community succeeds in making the procedure more consistent and accurate while retaining conformal guarantees, it could establish a framework in which the predictive model is inherently equipped with uncertainty quantification - a property that is essential in domains such as medicine, where the need for reliable uncertainty estimation is known in advance, even before training the model.

# E. Datasets

Examples of images used for model training and evaluation are presented in figure 3. For the training set, we employ a set of augmentations to increase sample diversity and enhance model generalization. The standard pipeline consists

*Table 3.* Number of ID and OOD images in each dataset.

| Dataset | $ID_{train}$ | $ID_{test}$ | OOD | Total |
|---|---|---|---|---|
| Colored MNIST | 60,000 | 10,000 | 10,000 | 80,000 |
| MVTec | 1,319 | 326 | 635 | 2,280 |
| Stanford Dogs | 1,394 | 100 | 856 | 2,350 |
| Retinopathy | 28,086 | 7,022 | 865 | 35,973 |

of the following sequential operations:

1. **Resizing and Cropping:** Images are first resized so that their shorter edge is 256 pixels. A patch of $224 \times 224$ pixels is then randomly cropped from the image. For datasets with substantial object size variation, such as Stanford Dogs, a more aggressive `RandomResizedCrop` with a scale range from 0.3 to 1.0 of the original image area is applied.

2. **Geometric Transformations:** Random rotations (up to $\pm15$ degrees) and random horizontal flips are applied to encourage rotational and reflectional invariance.

3. **Photometric Transformations:** For datasets where color variations are informative, such as Stanford Dogs, `ColorJitter` is applied to randomly modify brightness, contrast, and saturation. Additionally, a small probability of converting the image to grayscale is included for the Stanford Dogs dataset.

4. **Tensor Conversion and Normalization:** Finally, the augmented image is normalized using the standard ImageNet per-channel mean and standard deviation.

For simpler datasets, such as CIFAR-10 and CIFAR-100 (Krizhevsky et al., 2009), a similar but smaller-scale pipeline is used, consisting of random horizontal flips and random crops on the original $32 \times 32$ images, followed by normalization using CIFAR-specific statistics.

The specific class compositions for the ID and OOD splits used in our experiments are detailed below.

**Colored MNIST**

- **ID Classes:** Digits 0–9, each correlated with a specific color.

- **OOD Data:** Test set digits with novel color-digit pairings.

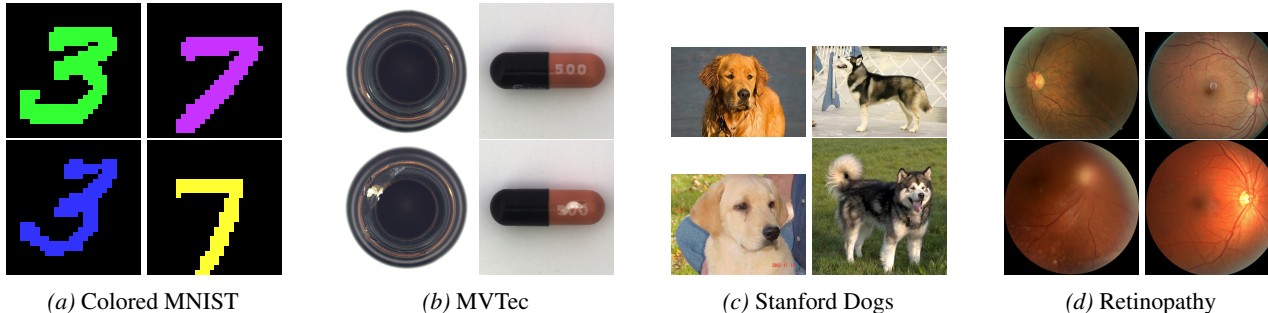

*(a)* Colored MNIST      *(b)* MVTec      *(c)* Stanford Dogs      *(d)* Retinopathy

*Figure 3.* Example images from four datasets. First row in each box: in-distribution; second row: outliers.

### Stanford Dogs

- **ID Classes (10):** Beagle, Boxer, Border Terrier, Dingo, German Shepherd, Giant Schnauzer, Golden Retriever, Old English Sheepdog, Siberian Husky, Standard Schnauzer.

- **OOD Classes (5):** Airedale, Labrador Retriever, Malamute, Malinois, Standard Poodle.

### MVTec AD

- **ID Classes (7 defect-free categories):** Bottle, Cable, Capsule, Metal Nut, Screw, Transistor, Zipper.

- **OOD Data (Anomalies):** Test set images of the same categories containing various defects.

### Retinopathy

- **ID Classes (5 DR severity levels):** No DR, Mild, Moderate, Severe, Proliferative.

- **OOD Data (Other Pathologies):** Fundus images depicting other diseases (e.g., Glaucoma, AMD, Pathological Myopia).

## F. Other Evaluations

### F.1. Near-OOD Datasets

We also evaluate our approaches on datasets that are commonly used in outlier exposure studies. Many previous works train classification models on datasets such as CIFAR-10 or CIFAR-100 and evaluate OOD detection on classical benchmarks like LSUN (Yu et al., 2015), Places365 (Zhou et al., 2017), and Textures (Cimpoi et al., 2014). A key limitation of this evaluation setting is that these datasets are primarily far-OOD: the ID data consist of categories such as cats, cars, and birds, while the OOD sets contain images of ground, bedrooms, towers, and similar content. Although this task is not necessarily easier, it is less representative of

practical scenarios. In contrast, the main part of this paper emphasizes near-OOD classes, which are more relevant for safety-critical AI systems.

Nevertheless, we also evaluated our methods on these widely used far-OOD datasets and compared GCOS not only with VOS but also with other methods from the literature. As shown in Tables 4, 5, and Tables 6, 7, our approaches achieve competitive performance relative to existing methods. We note that on this far-OOD regime GCOS is competitive rather than dominant: post-hoc inference-time methods such as ASH (Djurisic et al., 2023) achieve a higher Textures AUROC on CIFAR-100, and on Places365 Free Energy and VOS edge out GCOS.

### F.2. Multi-Seed Validation

We also evaluate GCOS performance across multiple random seeds (n=5 per dataset) showing mean $\pm$ standard deviation. Low variance across all metrics demonstrates hyperparameter robustness and result reproducibility; AUPR remains highly stable across all seeds. This consistency suggests that the method is not dependent on initialization lucky streaks.

### F.3. Object Detection

*Table 9.* OOD detection comparison on PASCAL-VOC (In-Distribution) with MS-COCO and OpenImages as OOD datasets.

| Method | FPR95 ↓ | AUROC ↑ | mAP (ID)↑ |
|---|---|---|---|
| | OOD: MS-COCO / OpenImages | | |
| MSP (Hendrycks & Gimpel, 2017) | 70.99 / 73.13 | 83.45 / 81.91 | 48.7 |
| ODIN (Liang et al., 2018) | 59.82 / 63.14 | 82.20 / 82.59 | 48.7 |
| Mahalanobis (Lee et al., 2018b) | 96.46 / 96.27 | 59.25 / 57.42 | 48.7 |
| Energy score (Liu et al., 2020) | 56.89 / 58.69 | 83.69 / 82.98 | 48.7 |
| Gram matrices (Sastry & Oore, 2020) | 62.75 / 67.42 | 79.88 / 77.62 | 48.7 |
| Generalized ODIN (Hsu et al., 2020) | 59.57 / 70.28 | 83.12 / 79.23 | 48.1 |
| CSI (Tack et al., 2020) | 59.91 / 57.41 | 81.83 / 82.95 | 48.1 |
| GAN-synthesis (Lee et al., 2018a) | 60.93 / 59.97 | 83.67 / 82.67 | 48.5 |
| VOS-ResNet50 | 47.53±2.9 / 51.33±1.6 | 88.70±1.2 / 85.23±0.6 | 48.9±0.2 |
| VOS-RegX4.0 | 47.77±1.1 / 48.33±1.6 | 89.00 ±0.4 / 87.59±0.2 | 51.6±0.1 |
| **GCOS (Ours)** | 51.28±<0.01 / 60.69±<0.01 | 87.72±<0 / 86.57±<0.01 | 48.8±<0.01 |

We also investigate how the GCOS method performs in object detection settings. As shown in Table 9, GCOS represents a significant advancement over traditional OOD detection methods, achieving 4-6 percentage point improve-

*Table 4.* Comparison of out-of-distribution detection performance trained on CIFAR-10.

| | | CIFAR-100 | | | LSUN-C | | | Places | | | Textures | | | Mean | | |
| | | AUPR | AUROC | FPR95 | AUPR | AUROC | FPR95 | AUPR | AUROC | FPR95 | AUPR | AUROC | FPR95 | AUPR | AUROC | FPR95 |
|---|---|---|---|---|---|---|---|---|---|---|---|---|---|---|---|---|
| Gaussian OOD | + $\mathcal{L}_{uncertainty}$ | **99.61** | **87.08** | **49.0** | **99.97** | 98.56 | **6.5** | 99.66 | 89.1 | **39.5** | 99.68 | 88.34 | **49.0** | 99.73 | 90.77 | **36.0** |
| GCOS OOD | + $\mathcal{L}_{uncertainty}$ | 99.48 | 83.63 | 56.5 | 99.97 | **98.74** | 7.0 | 99.6 | 88.36 | 42.5 | 99.48 | 82.39 | 60.5 | 99.64 | 88.28 | 41.63 |
| GCOS OOD | + $\mathcal{L}_{reg}$ | 99.54 | 82.48 | 72.5 | 99.96 | 98.12 | 8.5 | **99.82** | **92.3** | 41.5 | **99.76** | **90.19** | 50.5 | **99.77** | **90.77** | 43.25 |

*Table 5.* Comparison of out-of-distribution detection performance trained on CIFAR-100.

| | | CIFAR-10 | | | LSUN-C | | | Places | | | Textures | | | Mean | | |
| | | AUPR | AUROC | FPR95 | AUPR | AUROC | FPR95 | AUPR | AUROC | FPR95 | AUPR | AUROC | FPR95 | AUPR | AUROC | FPR95 |
|---|---|---|---|---|---|---|---|---|---|---|---|---|---|---|---|---|
| Gaussian OOD | + $\mathcal{L}_{uncertainty}$ | **99.35** | 76.47 | 80.0 | 99.92 | 96.54 | 21.5 | 99.32 | 75.51 | 84.0 | **99.3** | 74.53 | 84.5 | 99.47 | 80.76 | 67.5 |
| GCOS OOD | + $\mathcal{L}_{uncertainty}$ | 99.33 | 76.13 | 83.5 | **99.94** | **97.1** | **17.0** | 99.37 | **77.18** | 81.0 | 99.23 | 72.96 | 87.5 | **99.47** | **80.84** | **67.25** |
| GCOS OOD | + $\mathcal{L}_{reg}$ | 99.16 | 72.53 | 89.5 | 99.79 | 91.63 | 38.0 | 99.16 | 72.95 | 84.5 | 99.3 | 76.79 | 78.0 | 99.35 | 78.47 | 72.5 |

*Table 6.* Results on CIFAR-10.

| | Textures | | Places365 | |
| | AUROC | FPR95 | AUROC | FPR95 |
|---|---|---|---|---|
| Free Energy (Liu et al., 2020) | 85.35 | 52.46 | 90.02 | 40.11 |
| ASH (Djurisic et al., 2023) | 86.07 | 50.90 | 89.79 | 40.89 |
| VOS (Du et al., 2022) | 88.34 | 49.00 | 89.10 | 39.50 |
| GCOS + $\mathcal{L}_{reg}$ | 90.19 | 50.50 | 92.3 | 41.50 |

*Table 7.* Results on CIFAR-100.

| | Textures | | Places365 | |
| | AUROC | FPR95 | AUROC | FPR95 |
|---|---|---|---|---|
| Free Energy (Liu et al., 2020) | 76.35 | 79.63 | 75.65 | 80.18 |
| ASH (Djurisic et al., 2023) | 83.59 | 63.69 | 74.87 | 79.70 |
| VOS (Du et al., 2022) | 74.53 | 84.50 | 75.51 | 84.00 |
| GCOS + $\mathcal{L}_{reg}$ | 76.79 | 78.00 | 72.95 | 84.50 |

*Table 8.* Multi-Seed Validation.

| Dataset | Mean AUROC | Mean AUPR | Mean FPR95 | Std(AUROC) | Std(AUPR) |
|---|---|---|---|---|---|
| Colored MNIST | 92.90% | 99.81% | 23.50% | ±3.01% | ±0.10% |
| Stanford Dogs | 97.39% | 99.51% | 24.00% | ±2.11% | ±0.39% |
| Retinopathy | 76.60% | 99.05% | 80.00% | ±0.55% | ±0.02% |
| MVTec | 98.71% | 99.66% | 3.08% | ±0.29% | ±0.12% |

lanobis scoring. In the main classification setup, WRN-40-2 uses 128-dimensional penultimate features, so these computations are much smaller than image-space generation and reflect a categorically different cost profile from diffusion-based methods (Dream-OOD, NCIS), which require full image-space generation.

We report a microbenchmark of GCOS overhead on a single NVIDIA RTX 6000 Ada Generation GPU, averaged over 100 iterations and using synthetic feature queues that match training defaults (`feat_dim`=128, `queue_size`=1000, `synthesis_per_class`=10, `num_directions`=2, `binary_search_steps`=15, `variance_threshold`=0.90). PCA decomposition and shell calibration run once per epoch and are amortised across the ∼1,600 batches of a typical epoch at ImageNet-100 scale, leaving the binary-search synthesis step as the dominant per-batch cost. All three phases scale linearly with the number of classes $K$.

*Table 10.* Per-batch GCOS overhead by number of classes $K$. NVIDIA RTX 6000 Ada Generation; 100-iteration average.

| $K$ | PCA (ms) | Shell Calib (ms) | Synthesis (ms) | Total (ms) |
|---|---|---|---|---|
| 10 | 16.7 | 2.3 | 35.8 | 54.8 |
| 20 | 33.6 | 4.4 | 71.3 | 109.3 |
| 50 | 83.7 | 11.4 | 183.8 | 278.9 |
| 100 | 165.4 | 23.1 | 379.9 | 568.3 |
| 200 | 332.1 | 46.3 | 764.8 | 1143.3 |
| 1000 | 1663.7 | 239.6 | 3845.3 | 5748.6 |

ments in AUROC on both COCO and OpenImages OOD datasets. GCOS achieves competitive performance with the state-of-the-art VOS method, particularly excelling in AUROC on OpenImages (86.57 vs 85.23 for VOS-ResNet50) while maintaining comparable in-distribution mAP (48.8 vs 48.9), demonstrating that the method does not sacrifice primary task performance. The method successfully demonstrates that conformal prediction-based outlier synthesis can effectively improve OOD detection while maintaining strong in-distribution performance. GCOS shows robust performance across both OOD datasets, with standard deviation less than 0.01 across three independent training runs, indicating high reproducibility and stability.

In conclusion, the key contribution of GCOS is a principled, geometry-based approach to OOD detection that outperforms heuristic-based methods and competes with the best synthesis-based approaches across both classification and object detection scenarios.

## G. Computational Cost and Scalability

GCOS operates entirely in the penultimate feature space: the per-epoch calibration is a single forward pass plus per-class PCA decomposition, and the per-batch synthesis is a binary search along pre-computed directions plus Maha-

**Scalability to ImageNet-class settings.** To assess scalability beyond the CIFAR-class setting, we fine-tuned both GCOS and VOS on ImageNet-100 starting from the same ResNet-34 pretrained on ImageNet-1K. Both models were trained for 10 epochs under identical conditions, using the same default GCOS hyperparameters as the CIFAR experi-

| OOD Dataset | GCOS AUROC | VOS AUROC | GCOS FPR@95 | VOS FPR@95 |
|---|---|---|---|---|
| iNaturalist | **91.6** | 90.8 | **41.5** | 46.0 |
| SUN | **94.7** | 94.6 | **24.0** | 28.5 |
| Places | **95.0** | 94.7 | 22.5 | 22.5 |
| Textures | 95.9 | **96.4** | 22.5 | **16.5** |
| **Mean** | **94.3** | 94.1 | **27.6** | 28.4 |

ments with no ImageNet-specific tuning, and evaluated with energy scoring on the four standard OOD benchmarks.

GCOS improves over VOS on three of four datasets by AUROC and matches or improves FPR@95 on three of four; the gap is largest on iNaturalist ($+0.8$ pp AUROC, $-4.5$ pp FPR@95). Together with the CIFAR-100 100-class classification results and the cross-task PASCAL-VOC object-detection results in Table 9, the results indicate that GCOS continues to work at higher class counts and with a backbone (ResNet-34) different from the WRN-40-2 used in the main classification experiments.

## H. Score function and Loss Ablations

Throughout this work, we use the Mahalanobis distance to define the synthesis shell because its value reliably increases as a point moves away from a class mean along a principal component. In contrast, the Energy Strangeness score lacks this property at the beginning of training: its landscape can be complex and non-monotonic, so a larger geometric deviation $\alpha$ does not necessarily correspond to a higher, more OOD-like energy score, making it unsuitable for our boundary-finding search algorithm and thus for synthesis.

It is a notable finding that while optimizing for Mahalanobis geometric loss, $\mathcal{L}_{reg}$(Mahalanobis), it effectively regularizes the energy-based confidence of the classifier, the reverse is not true, as the non-monotonic nature of the energy landscape makes it an unreliable guide for our geometric synthesis search. From Table 12, we observe high scores for the GCOS outliers + $\mathcal{L}_{reg}$(Mahalanobis) configuration, along with a slight improvement over the GCOS outliers + $\mathcal{L}_{\text{uncertainty}}$ uncertainty loss (2) for MVTec and Colored MNIST. This suggests a deep and powerful connection between the geometric structure of the feature space and the confidence landscape of the classifier's logits. This relationship may be explained as follows:

- **Geometric Compactness as a Regularizer.** Constraining feature clusters to remain geometrically compact through a Mahalanobis-based geometric loss encourages the network to place in-distribution class features on low-dimensional manifolds. This facilitates the final classifier layer's ability to assign high-confidence logits, leading to consistently low energy scores.

- **Coupling of Geometric and Confidence-Based**

**Strangeness.** A feature vector that is geometrically atypical - i.e., associated with a high Mahalanobis score - also tends to elicit classifier uncertainty, manifested as high energy. Since the classifier learns linear boundaries in feature space, points located far from their cluster are more likely to fall into ambiguous regions with weak logits. Consequently, optimizing geometric OOD scores based on Mahalanobis distance can effectively serve as a proxy for energy-based OOD optimization.

## I. Conformal Risk Control

**Note on Decision-Making with Conformal Risk Control.** In addition to the direct hypothesis test described in equation (7), the p-values can be converted into a continuous OOD score to enable explicit conformal risk control (Angelopoulos et al., 2024). We define the final OOD score for a test sample as $S_{OOD}(\mathbf{z}^{\text{test}}) = 1 - p_{\text{final}}(\mathbf{z}^{\text{test}})$, where higher values indicate a higher likelihood of being OOD. Using this score, a decision threshold $\tau$ can be chosen to control a desired error rate on the in-distribution data. For instance, in VOS, the logistic regression threshold was set so that 95% of ID data is labeled as in-distribution; here, we adopt a similar logic using p-value thresholding.

Specifically, to control the False Negative Rate - the rate at which ID samples are incorrectly classified as OOD-at a level of $\alpha_{risk}$, we may set $\tau$ to be the $(1 - \alpha_{risk})$-quantile of the OOD scores computed on the final calibration set $\mathcal{D}_{calib}$. A test sample is then classified as OOD if its OOD score exceeds $\tau$, and as ID otherwise.

This approach ensures that the FNR on future data is approximately controlled at $\alpha_{risk}$. While the direct p-value test enforces a fixed statistical significance level, the risk-controlled method provides flexibility by allowing threshold selection to achieve a specific desired performance on the ID data, such as the 95% True Positive Rate (corresponding to a 5% FNR) often used for computing FPR@95TPR.

## J. Hyperparameter Sensitivity

Our approach introduces several hyperparameters that influence model performance across different settings, as illustrated in figure 4. The hyperparameters appear to exhibit synergies and correlated effects on performance.

The number of directions denotes the number of vectors in $\tilde{V} \subseteq \mathbf{V}_{\text{small, train}}^k$, a random subsample of $\mathbf{V}_{\text{small, train}}^k$. For the *avg direction* configuration, a single outlier is generated along the direction of the average vector $v$ as in (3), whereas *per direction* generates $|\tilde{V}|$ OOD vectors. The effect of the number of directions (first row of the figure 4) depends on the values of other hyperparameters. For *avg direction*

*Table 12.* **Ablation study of proposed contributions**. OOD detection results on MVTec, Colored MNIST, Stanford Dogs and Retinopathy datasets. First row: GCOS outliers combined with the standard energy-based separation loss $\mathcal{L}_{\text{uncertainty}}$ in (2). Second row: GCOS outliers combined with the geometric loss $\mathcal{L}_{\text{reg}}$ in (5) using the Mahalanobis score function $\mathcal{S}_{\mathcal{L}} = \mathcal{S}_{\text{Mahal}}$.

| | MVTec | | | Colored MNIST | | | Stanford Dogs | | | Retinopathy | | |
| --- | --- | --- | --- | --- | --- | --- | --- | --- | --- | --- | --- | --- |
| | AUROC | AUPR | FPR95 | AUROC | AUPR | FPR95 | AUROC | AUPR | FPR95 | AUROC | AUPR | FPR95 |
| GCOS + $\mathcal{L}_{\text{uncertainty}}$ | 93.07 | 98.56 | 33.85 | 93.21 | 99.82 | 25.50 | **99.42** | **99.89** | **0.86** | **80.05** | **99.23** | **71.00** |
| GCOS + $\mathcal{L}_{reg}$(Mahalanobis) | **97.56** | **99.51** | **20.00** | **95.65** | **99.91** | **25.00** | 97.90 | 99.61 | 20.00 | 77.89 | 99.17 | 79.00 |

*Figure 4.* Hyperparameter ablation study.

(columns 1 and 3), increasing *num directions* consistently improves AUROC and decreases FPR95, suggesting that averaging more diverse directions produces better and more informative outliers.

Shared Covariance controls the estimation strategy for the covariance matrix of the "Proposer" PCA models. When enabled, a single shared covariance matrix is estimated from the pooled, class-centered features of all categories in the training queue, under the assumption of a common underlying geometric structure. When disabled, a distinct per-class covariance is estimated, allowing for more specific manifold modeling at the expense of statistical robustness for classes with fewer samples. Its impact appears most notable for very small values of *num directions*; beyond this regime, its effect on performance is minimal.

The value of *synthesis per class* (second row of the figure 4) dictates the number of virtual outliers generated for each

class per training iteration. It directly controls the cardinality of the synthetic OOD set, $\mathcal{Z}_{ood}$, used to compute the regularization loss, thereby modulating the strength of the outlier exposure signal in each optimization step. The effect of this hyperparameter depends on the values of other hyperparameters, with no clear trend observed, suggesting that the total number of outliers may be less important than their quality and diversity.

The variance threshold (third row of the figure 4), $\eta \in (0, 1)$, determines the partitioning of the principal components of the "Proposer" models into a high-variance signal subspace, $\mathbf{V}_{\text{large}}$, and a low-variance off-manifold subspace, $\mathbf{V}_{\text{small}}$. Components are classified according to the minimum number of principal components required to explain at least $\eta$ proportion of the total feature variance; the remaining components form $\mathbf{V}_{\text{small}}$ and serve as directions for outlier synthesis. A value of 0.9 appears to provide the best perfor-

mance, being sufficiently permissive to include more components in $\mathbf{V}_{\text{small, train}}^{k}$, which allows for more diverse and flexible outlier generation. As the threshold approaches 1.0, performance tends to decline, indicating that a more conservative definition of off-manifold directions (e.g., $\eta = 0.99$) leaves too few directions in $\mathbf{V}_{\text{small, train}}^{k}$ for effective synthesis. This highlights the importance of aggressively defining off-manifold directions.

