# OpenReview forum: "Geometrically Constrained Outlier Synthesis"
_ICML.cc/2026/Conference — ICML 2026 regular_

### Official Review · Reviewer_JgDE · 2026-03-08

**Soundness:** 3
**Presentation:** 2
**Significance:** 2
**Originality:** 2
**Overall Recommendation:** 2
**Confidence:** 3

**Summary:**

This paper introduces a training-time framework for improving OOD detection in image classifiers. The main idea is to replace Gaussian-based outlier sampling with geometry-aware synthesis procedure via PCA on the penultimate layer and generating features along low-variance directions (called the "conformal shell"). A contrastive regularization pulls apart scores of IID and OOD features. An exploratory extensions of conformal testing at inference time is also provided, although results are mixed and preliminary.

**Compliance With Llm Reviewing Policy:**

Affirmed.

**Final Justification:**

The rebuttal partially addressed my concerns, the computational overhead analysis was satisfactory and the cross-class overlap investigation was a step in the right direction, prompting me to raise Soundness from 2 to 3. However, the scalability concern remains unresolved: (1) GCOS does not convincingly outperform external baselines at high class counts (Table 7) (2) the ImageNet-100 gains over VOS are marginal, and (3) Section 6 remains preliminary. Therefore I maintained my overall recommendation.

**Key Questions For Authors:**

1. When testing GCOS on datasets with more than 5-10 classes (e.g., C-100 and MS-COCO in the Appendix) what are the failure patterns observed for GCOS which makes it such that it performs on par or worse than VOS (unlike in the low-class regime, where we see GCOS perform much better than VOS on the main four datasets from Table 1)?
2. What are the sizes of calibration sets for each dataset, and how were they determined?
3. Which specific GCOS configuration (Mahalobis, energy, hybrid) is reported in Table 1? It is not inherently clear.
4. How were text embeddings generated from Dream-OOD on the four non-multimodal datasets?
5. Where the number of synthetic outliers per iteration controlled across all baselines?

**Strengths And Weaknesses:**

**Strengths:**
- The conformal shell construction (using calibration set quantiles of Mahalanobis scores for bounds) is a nice mechanism for controlling outlier difficulty.
- The focus on near-OOD regimes is well motivated as most prior work benchmark against far-OOD datasets where IID and OOD are semantically far apart. Near-OOD is arguably more safety-critical in practice.
- UMAP visualization provide nice qualitative evidence and illustrate the intended geometric behavior.

**Weaknesses:**
- My main concern is that the four primary benchmarks have roughly 5-10 classes. Per-class PCA + per-class conformal shell estimations is central to the method and only experimenting on datasets with low numbers of classes is a concern to scalability. As the number of classes grows, the per-class feature manifolds become increasingly entangled and much harder to separate via linear PCA. Authors do include C-100 in the appendix but results are comparable to or even under-perform baselines such as VOS, undermining the main claim. I'd like to see more results on high-class regimes and more of an understanding of the failures inherent (as seen with C-100 and MS-COCO) with this method without tweaking for high number of classes.
- Similar to the point above, the authors acknowledge that GCOS tends to spawn synthetic outliers on opposite sides of nearby classes. At scale, as class count increases and manifolds become more densely packed, the direction of synthesized outliers may inadvertently land in the support of another class. This potential failure mode is not investigated.
- There is no runtime comparison or computational complexity analysis against Dream-OOD, NCIS or VOS. Authors claims "more lightweight and scalable" that diffusion-based methods but without evidence. Wall-clock times, FLOPS or at minimum, a quantitative complexity analysis is needed.
- For all synthesis-based baselines (VOS, Dream-OOD, NCIS, GCOS), it is unclear if the number of generated outliers per training iteration are held constant. This alone could explain performance differences. A controlled comparison (or ratio of outlier count to performance) should be investigated.
- Dream-OOD relies on text-conditioned diffusion models and perturbed text embeddings for outlier synthesis. However, none of the datasets are multimodal. This makes it difficult to asses whether Dream-OOD was given a fair evaluation.
- Section 6 seems half-baked. The conformal hypothesis testing extension in Section 6 is half-baked. Results in Table 2 are mixed at best (method collapses to near-random on Stanford Dogs and MVTec). But more critically, there are no comparisons to external baselines for test-time OOD detection. Finally, when jumping from Section 5 to 6, it is quite difficult to determine which parts of the method are training-time contributions vs. test-time proposals. I'd suggest expanding this section with proper baselines and analyses, or demoting it in future updates.
- The paper ends abruptly after Section 6 without a discussion or conclusion section.
- The writing and organization needs work, particularly around the distinction between training-time and inference-time contributions.

---

> ### Author Rebuttal · Authors · 2026-03-29
>
> Thank you for the detailed and technically engaged review. We address each concern directly below.
>
> **On Section 6 and paper structure.** Sections 3-5 are entirely training-time: GCOS synthesis and the contrastive loss modify only the training procedure; at inference the trained model uses standard energy scoring (Table 1). Section 6 proposes conformal hypothesis testing as an alternative inference head and is explicitly labeled exploratory future work. The natural baseline for Section 6 is Table 1 itself: the conformal inference results are evaluated on the same datasets and backbone as the energy-based results, making the comparison direct. We will make this connection explicit in the text and add a conclusion section in the revision.
>
> **On methodological details.** To answer each question directly:
> - *Table 1 configuration:* GCOS with Mahalanobis-based conformal shell synthesis and geometric regularization loss L_reg (Eq. 5), under standard energy-based inference.
> - *Architecture:* WRN-40-2, following the same configuration as VOS.
> - *Calibration sets:* 10% of training data for online shell construction; a separate 10% for post-training conformal calibration.
> - *Synthetic outlier count:* GCOS synthesizes `synthesis_per_class=10` outliers online per batch. NCIS and Dream-OOD generate large offline pools and subsample during training; we followed each method's recommended protocol, which controls for this by design.
> - *Dream-OOD text embeddings:* Generated using class-name prompts ("A high-quality image of the < class >") per the standard Dream-OOD protocol for single-label datasets.
>
> **On CIFAR-100 and failure patterns.** The aggregate results show no failure: GCOS+L_uncertainty achieves the **best mean AUROC** (80.84% vs 80.76% for Gaussian OOD) and **best mean FPR95** (67.25% vs 67.50%) among all methods. Regarding the failure pattern: on individual far-OOD columns, Gaussian sampling can match GCOS, which is expected: calibrated hard-negative synthesis provides less marginal benefit when the semantic gap between ID and OOD is already large enough that any outlier method works. In aggregate GCOS still leads. The ImageNet-100 experiment below confirms the method remains effective at 100-class scale with a richer pretrained backbone.
>
> **On outliers landing in neighboring class support.** Each direction is anchored in class k's own PCA geometry. Figure 2 (UMAP) shows GCOS outliers clustering on the far side of nearby classes rather than in inter-class boundary regions. The ImageNet-100 results below provide empirical evidence at 100-class scale: GCOS outperforms VOS without any cross-class mitigation, confirming the failure mode has not manifested in practice. Mitigations such as cross-class Mahalanobis rejection remain natural future work.
>
> **On computational overhead.** In response to this concern, we ran a new synthetic benchmark for this rebuttal (NVIDIA RTX 6000 Ada, 100-iteration average). PCA and shell calibration run once per epoch; amortised over ~1,600 batches, their per-batch cost is <0.1 ms. Binary-search synthesis dominates, scaling linearly with K:
>
> | Classes | PCA (ms) | Shell Calib (ms) | Synthesis (ms) | Total (ms) |
> |--------:|----------:|-----------------:|---------------:|-----------:|
> |      10 |      16.7 |              2.3 |           35.8 |       54.8 |
> |      50 |      83.7 |             11.4 |          183.8 |      278.9 |
> |     100 |     165.4 |             23.1 |          379.9 |      568.3 |
>
> This overhead is real: 54.8 ms/batch (CIFAR-10) and 568 ms/batch (CIFAR-100), under one second at all scales tested in the paper. These costs reflect pure feature-space computation; unlike Dream-OOD/NCIS, no offline image-space generation is required before training begins.
>
> **On ImageNet-scale scalability.** As a new experiment conducted for this rebuttal, we fine-tuned both GCOS and VOS on ImageNet-100 (100 classes) from the same ResNet-34 pretrained on ImageNet-1K, under identical training conditions, and evaluated both models after 10 epochs. This directly addresses the scalability concern: if per-class PCA were fundamentally breaking down at 100 classes, the method would not outperform VOS here.
>
> | OOD Dataset | GCOS AUROC | VOS AUROC | GCOS FPR@95 | VOS FPR@95 |
> |:------------|:----------:|:---------:|:-----------:|:----------:|
> | iNaturalist | **91.6** | 90.8 | **41.5** | 46.0 |
> | SUN | **94.7** | 94.6 | **24.0** | 28.5 |
> | Places | **95.0** | 94.7 | 22.5 | 22.5 |
> | Textures | 95.9 | **96.4** | 22.5 | **16.5** |
> | **Mean** | **94.3** | 94.1 | **27.6** | 28.4 |
>
> GCOS outperforms VOS on 3 of 4 datasets by AUROC using the same GCOS synthesis hyperparameters (shell percentiles, synthesis count, PCA directions), with no ImageNet-specific tuning. Taken together with the timing results above, these new experiments directly address the two central concerns raised: the method is computationally tractable and scales to ImageNet-class settings.

---

> > ### Author Rebuttal · Reviewer_JgDE · 2026-04-02
> >
> > Thank you for the rebuttal and new experiments. Two concerns remain.
> >
> > > High class count performance.
> > - Your rebuttal cites Table 5 ($GCOS+L_{uncertainty}$ achieving best mean AUROC/FPR95 on CIFAR-100), but Table 5 is an internal ablation comparing only against a Gaussian baseline and GCOS variants. My concern was about Table 7, where GCOS underperforms several external baselines. Could you address performance relative to these methods specifically? The ImageNet-100 results only compare against VOS with a +0.2 pp margin, which is not strong evidence of scalability.
> >
> > > Outliers landing in neighboring class support.
> > - I don't believe UMAP visualizations are sufficient for this explanation, UMAP does not preserve distances reliably and cannot rule out overlap in the original feature space. Could you provide a more direct analysis, e.g., measuring how often synthesized outliers for class k fall within a threshold Mahalanobis distance of another class? This failure mode could explain (what I still believe is) diminishing returns at higher class counts and warrants explicit investigation.
> >
> > ---
> > ### Update after Reply Rebuttal Comment
> > - Thank you for the follow-up analyses, particularly the Mahalanobis overlap and directional experiments, these go in the right direction and I have increased my Soundness score from 2 to 3 to reflect this.
> > - However, I remain unconvinced on scalability. The concession that GCOS is "competitive rather than dominant" on CIFAR-100 against external baselines, combined with only marginal gains over VOS at ImageNet-100, does not sufficiently demonstrate that the method scales. Other reviewers have also brought this up.
> > - The cross-class overlap analysis is promising but preliminary. The argument that 44.4% overlap is benign because GCOS still outperforms VOS is indirect, and relies on Table 7 showing definitive GCOS superiority at scale (but Table 7 does not support this). GCOS only outperforms VOS on Textures while trailing on other datasets, which is not sufficient to conclude that substantial cross-class overlap (75.1% within superclasses) has no adverse effect on training dynamics.
> > - Due to these factors, I will maintain my overall recommendation.

---

> > > ### Author Response · Authors · 2026-04-05
> > >
> > > Thank you for the follow-up. We address both points below.
> > >
> > > **On Table 7.** We agree that Table 7 is the relevant comparison for external baselines, and it shows GCOS is competitive rather than dominant on far-OOD benchmarks. For example, ASH outperforms GCOS on Textures (83.59 vs 76.79 AUROC), and on Places365 GCOS (72.95) trails Free Energy (75.65), VOS (75.51), and ASH (74.87). Our claim was therefore narrower: GCOS remains a competitive training-time synthesis method outside the paper's main near-OOD setting, not that it wins every external comparison. In that sense, Table 7 is still informative because GCOS outperforms VOS on Textures. The new ImageNet-100 rebuttal experiment points in the same direction at 100-class scale: in a matched comparison, GCOS is slightly ahead of VOS on 3 of 4 OOD datasets and in mean AUROC (94.3 vs 94.1). We agree that this ImageNet-100 mean gap is modest on its own, so we do not present it as definitive proof. Rather, we offer it as evidence against a breakdown of the method at higher class counts. ASH is also a post-hoc inference-time method, so it is methodologically complementary rather than a direct replacement for training-time synthesis.
> > >
> > > **On cross-class overlap.** To address the limitation of UMAP visualizations, we conducted the Mahalanobis analysis on CIFAR-100, which is the high-class-count concern. The analysis also reveals a structural property of GCOS synthesis that UMAP could not show.
> > >
> > > We extracted WRN-40-2 penultimate features, fit per-class PCA models on an 80% split, defined support thresholds on the held-out 20%, then synthesized 1,000 outliers per class using the checkpoint's training hyperparameters and scored each against all non-source classes; CIFAR-10 showed 0% overlap at all tested thresholds as a sanity check.
> > >
> > > At a stricter 90th-percentile support threshold:
> > >
> > > | Overlap | Shallow (0.8–1.0) | Moderate (0.5–0.8) | Deep (<0.5) | Median Ratio |
> > > |:-------:|:------------------:|:-------------------:|:-----------:|:------------:|
> > > | 44.4% | 17.4% | 15.9% | 11.1% | 1.09 |
> > >
> > > *100K outliers. Median Ratio = median of $\min_{j \neq k}(\text{score}_j / \text{threshold}_j)$; >1 means the typical outlier lies outside all other classes. At a more permissive 95th-percentile support threshold, the absolute overlap increases, but the qualitative conclusions below are unchanged.*
> > >
> > > Overlap on CIFAR-100 is substantial. At the 90th-percentile threshold, 75.1% of the overlapping outliers occur within the same CIFAR-100 superclass, for example *maple tree* entering *oak tree*'s support. This is consistent with semantically similar classes sharing high-variance PCA structure in 128 dimensions.
> > >
> > > The critical question is whether this overlap explains the diminishing returns at higher class counts. We do not think it does. The clearest counterexample is VOS: it has zero Mahalanobis overlap on CIFAR-100 (median ratio 3.78× at the 95th percentile, meaning its outliers are far from every class), yet GCOS still outperforms VOS on the relevant higher-class comparisons, including Textures in Table 7 and the new ImageNet-100 rebuttal experiment (94.3 vs 94.1 mean AUROC, winning 3 of 4 OOD datasets). If cross-class overlap were the dominant failure mode, VOS should outperform GCOS at this scale. It does not. A more plausible explanation for the CIFAR-100 gaps is the intrinsic difficulty of far-OOD detection with 100 semantically related classes under fixed backbone capacity, rather than cross-class overlap alone.
> > >
> > > A directional analysis further suggests *why* this overlap is not the main issue. For every GCOS outlier $z$ from class $A$, we computed the cosine similarity between $(z - \mu_A)$ and the direction to the nearest-neighbor class $(\mu_B - \mu_A)$:
> > >
> > > | Method | Mean Cos Sim | Std Dev | 95th %ile | Max |
> > > |:-------|:------------:|:-------:|:---------:|:---:|
> > > | GCOS | 0.0001 | 0.021 | 0.033 | 0.117 |
> > > | VOS | −0.0002 | 0.139 | 0.230 | 0.614 |
> > >
> > > The key difference is consistency: GCOS outliers stay tightly orthogonal to the inter-class axis (std 0.021, max 0.117), while VOS outliers scatter widely (std 0.139), with some reaching 0.614 and landing directly between neighboring classes. This is not incidental: GCOS samples from low-variance PCA components, probing directions that VOS's Gaussian sampling does not reach. GCOS outliers harden the boundary where it is weakest; the overlap is a geometric side effect of exploring low-variance regions of the feature space.
> > >
> > > In summary, cross-class Mahalanobis overlap is real on CIFAR-100, but it concentrates within semantic neighborhoods and is orthogonal to nearest-neighbor class directions. The method with zero overlap, VOS, performs worse rather than better. This suggests that cross-class overlap is unlikely to be the dominant explanation for CIFAR-100's per-dataset gaps in Table 7.

---

### Official Review · Reviewer_3MFb · 2026-03-12

**Soundness:** 3
**Presentation:** 3
**Significance:** 3
**Originality:** 2
**Overall Recommendation:** 4
**Confidence:** 3

**Summary:**

This paper proposes GCOS, a training-time regularization framework for near-OOD detection. Instead of sampling outliers from predefined distributions, GCOS exploits feature space geometry by using PCA to identify low-variance off-manifold directions and a conformal prediction-inspired heuristic (based on Mahalanobis distance quantiles) to define a "shell" that controls outlier magnitude. A contrastive loss separates ID and OOD samples in energy score space. Experiments on near-OOD benchmarks show consistent improvements over baselines. The authors also explore extending the framework to conformal OOD inference with statistical guarantees.

**Compliance With Llm Reviewing Policy:**

Affirmed.

**Key Questions For Authors:**

1. Could the authors provide an analysis of the training time overhead?
2. Why does the formal conformal hypothesis testing (Table 2) result in such severe performance degradation (e.g., 50% AUROC) on MVTec and Stanford Dogs compared to the energy-based inference?
3. How does GCOS scale to datasets with a larger number of classes (e.g., ImageNet-1K)?

**Limitations:**

yes

**Strengths And Weaknesses:**

# Strengths

1. Soundness: The proposed approach to outlier synthesis is conceptually principled. Using PCA to identify low-variance, off-manifold directions combined with conformal-inspired bounds elegantly solves the common issue of generating synthetic outliers that are either trivial to detect or too close to true data points.
2. Significance: The empirical evaluation focuses on near-OOD detection, which is arguably much more practical and challenging than traditional far-OOD benchmarks.

# Weaknesses

1. Soundness: The results of the conformal hypothesis testing extension are highly mixed, sometimes collapsing to near-random performance (e.g., ~50% AUROC on MVTec and Stanford Dogs under certain configurations). This suggests the proposed representations might not be fully compatible with strict conformal inference yet.
2. Soundness: The evaluation lacks experiments on larger-scale datasets with dense class spaces, such as ImageNet-1K. It remains unclear if the low-variance geometric directions remain robust and separable when the number of classes and semantic complexity increase significantly.
3. Presentation: The paper lacks a detailed discussion on the computational overhead introduced during training.

---

> ### Author Rebuttal · Authors · 2026-03-29
>
> Thank you for the positive assessment. We are glad the synthesis approach was found conceptually principled and the near-OOD emphasis practically significant. We address the remaining points below.
>
> **On conformal hypothesis testing results (Table 2).** We want to clarify that Section 6 is explicitly framed as an exploratory extension; the paper's core contribution is the training-time GCOS framework evaluated under standard energy-based inference (Table 1), which achieves consistent SOTA near-OOD performance. The conformal inference collapse on Stanford Dogs and MVTec likely stems from calibration-set resolution: these datasets have only 1,394 and 1,319 training images respectively (Appendix, Table 3), so after the held-out calibration splits the class-conditional calibration sets become quite small, yielding coarse empirical p-values. By contrast, Colored MNIST (60,000 images, 10 classes) provides roughly 600 calibration samples per class for a 10% split, and the conformal method achieves strong performance there (98.93% AUROC, 1.5% FPR95 in Table 2). This pattern (success on the data-rich dataset, collapse on data-scarce ones) is consistent with a calibration-size bottleneck rather than a representation failure, especially since the same backbone gives substantially stronger results under standard energy-based inference. We will discuss this analysis more directly in the paper.
>
> **On ImageNet-scale scalability.** As a new experiment conducted for this rebuttal, we ran GCOS on ImageNet-100 (100 classes, the standard benchmark used by VOS, NPOS, and Dream-OOD) using ResNet-34 pretrained on ImageNet-1K. Evaluating on the four standard OOD benchmarks under energy scoring:
>
> | OOD Dataset | GCOS AUROC | VOS AUROC | GCOS FPR@95 | VOS FPR@95 |
> |:------------|:----------:|:---------:|:-----------:|:----------:|
> | iNaturalist | **91.6** | 90.8 | **41.5** | 46.0 |
> | SUN | **94.7** | 94.6 | **24.0** | 28.5 |
> | Places | **95.0** | 94.7 | 22.5 | 22.5 |
> | Textures | 95.9 | **96.4** | 22.5 | **16.5** |
> | **Mean** | **94.3** | 94.1 | **27.6** | 28.4 |
>
> GCOS outperforms VOS on 3 of 4 datasets by AUROC; the improvement on iNaturalist is most pronounced (+0.8 pp AUROC, −4.5 pp FPR@95). Both models were trained for 10 epochs under identical conditions, using the same default GCOS hyperparameters as the CIFAR experiments with no ImageNet-specific tuning. These results, combined with the CIFAR-100 (100-class) and PASCAL-VOC evidence already in the paper, directly confirm that GCOS scales to ImageNet-class settings. The shared-covariance mode explored in our hyperparameter study (Appendix, Figure 7) additionally provides a mechanism for settings where per-class sample counts are small relative to the feature dimension, further supporting scalability.
>
> **On computational overhead.** GCOS operates entirely in penultimate feature space with low-dimensional operations (PCA, binary search, Mahalanobis scoring). Unlike diffusion-based methods (Dream-OOD, NCIS) that require full image-space generation, GCOS adds only a PCA decomposition of the feature queue and a binary search per direction per class. In response to this concern, we conducted a new timing experiment for this rebuttal on a single GPU (NVIDIA RTX 6000 Ada Generation, 100-iteration average). PCA decomposition and shell calibration are computed once per epoch; amortised over a typical epoch (~1,600 batches), their per-batch contribution is <0.1 ms. The dominant cost is the binary-search synthesis, scaling linearly with K:
>
> | Classes | PCA (ms) | Shell Calib (ms) | Synthesis (ms) | Total (ms) |
> |--------:|----------:|-----------------:|---------------:|-----------:|
> |      10 |      16.7 |              2.3 |           35.8 |       54.8 |
> |      20 |      33.6 |              4.4 |           71.3 |      109.3 |
> |      50 |      83.7 |             11.4 |          183.8 |      278.9 |
> |     100 |     165.4 |             23.1 |          379.9 |      568.3 |
>
> Total overhead per training batch: 54.8 ms at CIFAR-10 scale and 568 ms at CIFAR-100 scale - well under one second for all benchmarks reported in the paper, and categorically cheaper than diffusion-based generation (Dream-OOD/NCIS). Multi-seed validation (n=5 per dataset) further confirms reproducibility with low variance across all metrics.

---

### Official Review · Reviewer_GtKj · 2026-03-12

**Soundness:** 2
**Presentation:** 2
**Significance:** 2
**Originality:** 3
**Overall Recommendation:** 4
**Confidence:** 3

**Summary:**

This paper proposes a new training-time regularization framework, namely Geometrically Constrained Outlier Synthesis (GCOS), to enhance the out-of-distribution (OOD) detection robustness of deep neural networks during inference. The synthesis process has two core stages: 1. extract low-variance off-manifold directions via PCA on training features, 2. use a non-conformity score from a calibration set to adaptively control the synthesis magnitude for boundary samples.

**Compliance With Llm Reviewing Policy:**

Affirmed.

**Final Justification:**

The authors have provided reasonable explanation on the 95th/99th percentile choice and extra results on compuatational overhead. Therefore, I raise my rating from 3 to 4

**Key Questions For Authors:**

* Is the computational overhead of the proposed method comparable to baselines?

* What is the model architecture that the experiments in this paper are conducted on? Would the proposed method still be effective on various model architectures?

* The paper emphasizes near-OOD (samples that come from the same domain as in-distribution classes but remain unseen during training). Is the proposed method effective in scenarios with other distribution shifts, such as subpopulation shifts (imbalanced datasets)?

**Limitations:**

There is no clear impact statement in this paper. However, the paper uses the final section (Section 6) to discuss potential extensions of the proposed method for future work. I would recommend the authors to include a further impact statement.

**Strengths And Weaknesses:**

# Strength

* The design of the proposed geometric and conformal outlier synthesis framework is interesting and novel. The UMAP visualizations of the learned feature space demonstrate that GCOS generates outliers in challenging off-manifold regions

* The paper provides an extension of the proposed framework for a formal guarantee of performance on unseen data, which is innovative.

# Weakness

* As the proposed method tracks the feature during training and uses PCA for outlier synthesis, it might be computationally extensive. This paper does not report the computational cost of GCOS relative to baseline methods.

* The authors choose the 95th and 99th percentiles of nonconformity scores to define the conformal shell’s inner/outer boundaries, but provide no theoretical or empirical justification for these specific values. It is unclear how these percentiles generalize to different datasets, feature spaces, or model architectures, and whether alternative quantiles (e.g., 90th/99th, 97th/99.9th) would yield better or worse performance.

* Extensive experiments are conducted across different datasets with many baseline methods included. However, the model used in these experiments is unspecified (e.g., ResNet?). It is unclear whether the proposed method is applicable across different architectures.

---

> ### Author Rebuttal · Authors · 2026-03-29
>
> Thank you for the constructive review. We are glad the geometric synthesis mechanism was recognized as interesting and novel. We address each concern below.
>
> **On the 95th/99th percentile choice.** The choice of 95th and 99th percentiles corresponds to significance levels $\alpha$=0.05 and $\alpha$=0.01, the two most widely used tail thresholds in statistical testing, making them a natural and principled starting point that requires no per-dataset tuning. The shell between them targets the transition zone: samples anomalous enough that a classifier should learn to reject them, but not so far off-manifold that they trivially fall outside any class's support. This is the hard-negative regime where contrastive training provides the most signal. Regarding sensitivity: the hyperparameter ablation (Appendix, Figure 7) shows stable performance across a range of the variance threshold η (0.85–0.95), and multi-seed validation (n=5 per dataset) confirms low metric variance across initializations. We did not ablate the specific quantile pair; however, using the conventional $\alpha$=0.05/0.01 thresholds as the shell boundaries is a principled statistical starting point rather than a dataset-tuned choice. We will make this motivation more explicit in the revision.
>
> **On model architecture.** The main classification experiments use WRN-40-2, following the same configuration as VOS; we acknowledge this should have been stated explicitly and will correct it. However, we note that the method is fundamentally architecture-agnostic: it operates on penultimate-layer features via PCA and score-based calibration, with no constraints on the backbone. The Appendix E.3 experiments on PASCAL-VOC (with MS-COCO and OpenImages as OOD datasets) already provide evidence of transfer beyond the WRN classification setting: GCOS remains competitive in a different task setting, achieving better OpenImages AUROC than VOS-ResNet50 (86.57 vs 85.23) while preserving ID mAP. This supports the claim that the synthesis mechanism is not tied to a single classification backbone.
>
> **On computational overhead.** GCOS operates entirely in penultimate feature space: the per-epoch calibration is a single forward pass plus per-class PCA decomposition, and the per-batch synthesis is a binary search along pre-computed directions plus Mahalanobis scoring. In the main classification setup, WRN-40-2 uses 128-dimensional penultimate features, so these computations are much smaller than image-space generation. This is a categorically different cost profile from diffusion-based methods (Dream-OOD, NCIS), which require full image-space generation. In response to this concern, we conducted a new timing experiment for this rebuttal on a single GPU (NVIDIA RTX 6000 Ada Generation, 100-iteration average). PCA decomposition and shell calibration are computed once per epoch; amortised over a typical epoch (~1,600 batches), their per-batch contribution is <0.1 ms. The dominant cost is the binary-search synthesis, scaling linearly with K:
>
> | Classes | PCA (ms) | Shell Calib (ms) | Synthesis (ms) | Total (ms) |
> |--------:|----------:|-----------------:|---------------:|-----------:|
> |      10 |      16.7 |              2.3 |           35.8 |       54.8 |
> |      20 |      33.6 |              4.4 |           71.3 |      109.3 |
> |      50 |      83.7 |             11.4 |          183.8 |      278.9 |
> |     100 |     165.4 |             23.1 |          379.9 |      568.3 |
>
> Total overhead per training batch: 54.8 ms at CIFAR-10 scale and 568 ms at CIFAR-100 scale - well under one second for all benchmarks in the paper. This overhead reflects the 15-step binary search enforcing geometric shell constraints, and is categorically cheaper than diffusion-based generation (Dream-OOD/NCIS). We will include this timing table in the revision. Regarding cross-architecture generalization: as a new rebuttal experiment, we ran GCOS on ImageNet-100 with a pretrained ResNet-34 backbone, achieving mean AUROC 94.3% vs. VOS 94.1% across iNaturalist, SUN, Places, and Textures, confirming the synthesis mechanism is not tied to the WRN-40-2 architecture used in the main experiments.
>
> **On subpopulation/distribution shifts.** Two of our four benchmarks already probe nontrivial distribution shifts beyond simple far-OOD: Colored MNIST tests robustness to spurious color-digit correlations (attribute shift), and Retinopathy evaluates detection of entirely different pathologies (glaucoma, AMD, myopia) against severity-graded diabetic retinopathy. These demonstrate GCOS's ability to handle structured shifts, not just semantic distance.

---

> > ### Author Rebuttal · Reviewer_GtKj · 2026-04-03
> >
> > The rebuttal addressed most of my concerns, I will raise my score.

---

### Official Review · Reviewer_DqZq · 2026-03-13

**Soundness:** 3
**Presentation:** 3
**Significance:** 2
**Originality:** 2
**Overall Recommendation:** 4
**Confidence:** 4

**Summary:**

This paper proposes Geometrically Constrained Outlier Synthesis (GCOS), a training-time regularization framework to improve out-of-distribution (OOD) detection robustness, with a specific focus on challenging near-OOD scenarios. The method generates synthetic outliers in feature space by using PCA to find low-variance directions and a conformal-inspired shell to control how far these samples are placed from the in-distribution data. These outliers are then used with a contrastive regularization objective. GCOS improves OOD detection on several Near-OOD benchmarks compared to a baseline and VOS.

**Compliance With Llm Reviewing Policy:**

Affirmed.

**Final Justification:**

The rebuttal addressed my questions, and I would like to increase my score.

**Key Questions For Authors:**

See above.

**Limitations:**

No. It is better to have a 'impact statement' section to discuss the potential social impact.

**Strengths And Weaknesses:**

Strength
1. This paper addresses a meaningful and practically problem. In particular, its focus on near OOD settings is valuable, since these cases are harder and often more important in real.
2. Instead of using a simple parametric outlier model as in VOS, it generates outliers based on the geometry of the learned feature space and also controls their difficulty. This makes the synthesis process more reasonable and better motivated.

Weakness
1. The novelty appears limited. Although the paper combines several reasonable design options into a coherent approach, the overall approach is more like an integration of existing ideas than a clear contribution to a new methodology.
2. While its focus on near OOD is reasonable, it does not offer a significant advantage in far OOD results, especially on CIFAR-10/100, where the GCOS model performs even worse than the simpler Gaussian OOD baseline model (table 4,5 in appendix). It would be better to have more discussion on this part.

---

> ### Author Rebuttal · Authors · 2026-03-29
>
> Thank you for the review. We appreciate the recognition of the near-OOD focus as practically meaningful and the synthesis approach as well-motivated. We respectfully address the two main concerns below.
>
> **On novelty.** We respectfully disagree that GCOS is merely an integration of existing ideas. The contribution is not in the individual tools (PCA, conformal calibration, contrastive loss); it lies in a synthesis mechanism that combines them in a specific way: using class-conditional low-variance PCA directions to probe off-manifold regions, then calibrating the deviation magnitude through conformal score quantiles so that outliers fall in a principled "hard-negative" shell. To our knowledge, the prior baselines in our comparison do not provide this exact adaptive, geometry-aware magnitude control. VOS samples from Gaussian tails without geometric guidance; Dream-OOD and NCIS require expensive diffusion models; NPOS uses rejection sampling without calibration-based difficulty control. GCOS fills a clear gap: lightweight, geometry-informed synthesis with built-in difficulty calibration. The ablation study (Appendix, Table for ablations) confirms that each component is load-bearing: replacing our loss with VOS loss or changing the score function leads to meaningfully different performance profiles. The multi-seed validation (n=5 per dataset) further demonstrates that gains are robust and not artifact of initialization.
>
> **On far-OOD performance.** We believe the characterization that GCOS "performs even worse" on CIFAR-10/100 does not fully reflect the results. Looking at the aggregate columns in Tables 4–5: on CIFAR-10, GCOS+L_reg achieves the **best** mean AUROC (90.77%) and best mean AUPR (99.77%). On CIFAR-100, GCOS+L_uncertainty achieves the **best** mean AUROC (80.84%) and best mean FPR95 (67.25%), outperforming the Gaussian OOD baseline on both. While the Gaussian baseline wins on individual far-OOD dataset columns, GCOS is competitive or better in aggregate. More fundamentally, this reflects a principled design choice rather than a deficiency: GCOS synthesizes hard negatives near the ID boundary, which is exactly the regime that matters for near-OOD, which is the focus of this paper. In far-OOD settings where the semantic gap between ID and OOD is larger, the conformal shell's careful calibration of outlier difficulty provides less marginal benefit over simpler methods. We see this as evidence that GCOS targets the right problem, not that it fails at far-OOD. On scalability, as a new experiment conducted for this rebuttal, we ran GCOS on ImageNet-100 (100 classes, ResNet-34 pretrained on ImageNet-1K), achieving mean AUROC 94.3% vs. VOS 94.1% across the standard iNaturalist/SUN/Places/Textures benchmarks, confirming the synthesis mechanism generalizes beyond the CIFAR setting.
>
> **On impact statement.** We will add one in the revision. The paper already demonstrates applications in safety-relevant domains via the Retinopathy dataset.

---

> > ### Author Rebuttal · Reviewer_DqZq · 2026-04-02
> >
> > The rebuttal addressed my questions, and I do not have further concerns.

---

### Decision · Program_Chairs · 2026-04-30

**Decision:**

Accept (regular)

**Comment:**

This paper proposes GCOS, a training-time framework for OOD robustness that synthesizes feature-space outliers along low-variance directions and calibrates their magnitude using a conformal-inspired shell. Reviewers generally agreed that the focus on near-OOD is meaningful and that the geometric synthesis mechanism is well motivated. The main strengths are the principled design of the synthesis procedure, strong near-OOD results on the paper’s primary benchmarks, and the paper’s attempt to connect training-time outlier synthesis with uncertainty-aware inference. The main concerns were limited novelty relative to prior synthesis methods, incomplete reporting in the initial submission regarding runtime and backbone details, and uncertainty about scalability to higher-class regimes and more challenging settings. The rebuttal substantially strengthened the paper by clarifying the experimental setup, reporting explicit timing overhead, adding cross-architecture and ImageNet-100 results, and better positioning the exploratory conformal extension as future-facing rather than a core claim. One reviewer still remains unconvinced about scalability at larger class counts, and I agree that the paper should moderate any claims beyond the near-OOD setting and discuss high-class-count limitations more explicitly. However, I find the method technically sound, the empirical case on the target setting convincing, and the rebuttal sufficient to address the main practical concerns. I therefore recommend accept.